# A disease-linked lncRNA mutation in RNase MRP inhibits ribosome synthesis

Nic Robertson[1], Vadim Shchepachev [2], David Wright[3], Tomasz W. Turowski [1], Christos Spanos [1], Aleksandra Helwak[1], Rose Zamoyska [3] & David Tollervey [1✉]

*RMRP* encodes a non-coding RNA forming the core of the RNase MRP ribonucleoprotein complex. Mutations cause Cartilage Hair Hypoplasia (CHH), characterized by skeletal abnormalities and impaired T cell activation. Yeast RNase MRP cleaves a specific site in the pre-ribosomal RNA (pre-rRNA) during ribosome synthesis. CRISPR-mediated disruption of *RMRP* in human cells lines caused growth arrest, with pre-rRNA accumulation. Here, we analyzed disease-relevant primary cells, showing that mutations in *RMRP* impair mouse T cell activation and delay pre-rRNA processing. Patient-derived human fibroblasts with CHH-linked mutations showed similar pre-rRNA processing delay. Human cells engineered with the most common CHH mutation ($70^{AG}$ in *RMRP*) show specifically impaired pre-rRNA processing, resulting in reduced mature rRNA and a reduced ratio of cytosolic to mito-chondrial ribosomes. Moreover, the $70^{AG}$ mutation caused a reduction in intact RNase MRP complexes. Together, these results indicate that CHH is a ribosomopathy.

[1] Wellcome Centre for Cell Biology, University of Edinburgh, Edinburgh, UK. [2] The Gurdon Institute and Department of Pathology, University of Cambridge, Cambridge, UK. [3] Ashworth Laboratories, Institute of Immunology and Infection Research, University of Edinburgh, Edinburgh, UK. ✉email: D.Tollervey@ed.ac.uk

RMRP mutations are associated with a spectrum of disorders presenting with skeletal dysplasia, abnormal hair, and immune deficiency with impaired T-cell activation[1]. The most common of these syndromes is called Cartilage Hair Hypoplasia (CHH). Patients with CHH have reduced life expectancy due to immune deficiency, but the mechanism underlying this problem is not understood[1]. Clinically, patients often experience recurrent infections, and also have a higher incidence of autoimmunity and cancer. At a cellular level, lymphocytes from CHH patients show reduced proliferation in response to stimulation and increased activation-induced cell death[2,3]. Naïve T cells are small and quiescent, but on activation increase in size over 24 h before beginning to rapidly divide[4,5]. This activation program is accompanied by a 10-fold increase in per-cell ribosome abundance, achieved through upregulation of both ribosomal protein (RP) and ribosomal RNA (rRNA) production[6,7].

Human rRNAs are transcribed by RNA polymerase I (RNAPI) as a long precursor, 47 S pre-rRNA (Supplementary Fig. 1). This polycistronic transcript includes the 18 S rRNA, destined for the small ribosomal subunit (SSU), and the 5.8 S and 28 S rRNA components of the large subunit (LSU)[8]. The rRNA sequences are flanked by 5′ and 3′ external transcribed spacers (5′ETS and 3′ETS) and separated by internal transcribed spacers (ITS1 and ITS2). A third LSU rRNA, 5 S, is transcribed separately by RNA polymerase III. During ribosome synthesis, the 47 S primary transcript undergoes a complex sequence of endonuclease cleavages and exonuclease digestion steps that remove the spacer regions. These process the 47 S RNA through a series of discrete pre-rRNA processing intermediates (Supplementary Fig. 1), to generate the mature rRNAs (Fig. 1A). Notably, human pre-rRNAs cannot be assigned to a single linear order, and there are at least two major alternative pathways. The existence of partially redundant pathways may enhance the overall efficiency and resilience of the system.

During transcription and processing, around 80 ribosomal proteins will assemble with the pre-rRNA, assisted by more than 200 protein assembly factors and over 100 small non-coding RNAs termed small nucleolar RNAs (snoRNAs) (reviewed in ref. [9]). Most snoRNAs function as guides for pre-rRNA nucleotide modification (methylation, acetylation, or pseudouridine formation) or as chaperones for pre-rRNA folding (reviewed in[10]). All known snoRNAs fall into two large families (termed boxC/D and boxH/ACA snoRNAs), with the exception of RNase MRP.

The human RMRP gene encodes a noncoding RNA, which associates with around 10 proteins in the RNase MRP complex[11,12]. All of these proteins are shared with the RNase P complex, which processes pre-tRNAs and includes the evolutionarily related ncRNA RPPH1[9]. RNase MRP was initially proposed to function in cleavage of an RNA primer during mouse mitochondrial DNA replication, giving its name: mitochondrial RNA processing[12,13]. However, in budding yeast, RNase MRP is required for cleavage at a specific site, designated A3, in the ITS1 region of the pre-rRNA. A3 cleavage provides an entry site for the 5′ exonuclease Rat1 (Xrn2 in humans) that then generates the 5′ end of the major form of 5.8 S rRNA[11,14–16]. Loss of MRP activity does not fully inhibit pre-rRNA processing due to the presence of a poorly understood alternative processing pathway. Mature 5.8 S rRNA is present in two different forms in all studied animals, fungi and plants[17–19]. In S.cerevisiae about 80% of 5.8 S is the short form ($5.8S_S$) generated by the MRP-dependent pathway cleavage[14,15,17]. The MRP-independent pathway generates $5.8S_L$, which is 5′ extended into ITS1 by 7 or 8 nucleotides. In consequence, loss of MRP activity in yeast results in a characteristic reduction in $5.8S_S$ relative to $5.8S_L$.

Previous analyses of fibroblasts from CHH patients showed increased abundance of species with a 5′ extension into ITS1[20].

By analogy with observations in yeast, this was predicted to reflect 5′ extension the 5.8 S rRNA. CRISPR-mediated disruption of RMRP in human HEK293 cells caused pre-rRNA accumulation consistent with defective ITS1 processing at "site 2" analogous to site A3 in yeast[21]. However, another analysis failed to observe an ITS1 processing defect at this site following RMRP depletion[22].

Several different human diseases, collectively termed ribosomopathies, are caused by mutations or haploinsufficiency in the RNAPI transcription machinery, pre-ribosome maturation factors or ribosomal proteins (reviewed in refs. [23–25]). Despite their function in all cells, ribosomopathies are generally associated with quite cell-type specific diseases, most commonly hemopoietic defects (reviewed in refs. [26,27]). All known ribosomopathies cause only mild deficits in the accumulation of cytoplasmic ribosomes, presumably because severe defects would be lethal in early development. In consequence, tissue specificity may be caused by the combination of a generally mild deficit in translation capacity, combined with tissue-specific exacerbating factors (perhaps high demand for protein synthesis combined with reduced activity in compensating pathways).

In this study we show that mutations in RMRP impair mouse T-cell activation and delay pre-rRNA processing, a phenotype recapitulated in patient-derived human fibroblasts. In cells engineered with the most common CHH mutation, we also observed changes in RNase MRP structure and reduction in the production of cytoplasmic ribosomes, establishing the defect as a ribosomopathy.

## Results

**Disruption of RMRP impairs T cell proliferation and rRNA processing.** Activating T cells require high rates of ribosome synthesis to support rapid cell division[6]. We hypothesized that this might confer particular vulnerability to partial disruption of RMRP function if its primary biological role is in pre-rRNA processing. To test this, we used T cells from OT-1 Rag[-/-] mice, all of which recognize the same ovalbumin-derived antigen. This system allows T cells to be activated in a relatively physiological manner, by adding Ova peptide to the culture medium. Naïve primary transgenic T cells were isolated, stained with a division tracker dye (CellTrace Violet), and activated by culturing with ovalbumin peptide (Fig. 1a). After 24 h, we disrupted RMRP by electroporating these cells with CRISPR components, including one of four guides targeting RMRP, or a no-guide control (mock; Supplementary Fig. 2a). Cells were then left to proliferate for a further 48 h.

CRISRP RNAs guide associated CAS proteins to generate site-specific double-strand breaks in the DNA. In the absence of a repair template, these can be repaired by an error-prone nonhomologous end-joining process that generates a high frequency of nucleotide insertion and deletions (indels) at the repaired break site. Guide efficiency was therefore estimated by calculating the proportion of alleles in the population containing an indel, using the Synthego ICE tool to deconvolute Sanger sequencing traces obtained from bulk populations[28]. This showed that guides 1 and 2 disrupted about 60% of alleles, whereas guides 3 and 4 were less efficient, mutating approximately 20% and 10%, respectively (Fig. 1b). No mutations were detected in the control. Proliferation after 48 h was assessed by flow cytometry using a division tracker dye, CellTrace Violet, which halves in fluorescent intensity each division (Fig. 1c,d). This showed that disruption of RMRP markedly impaired T cell proliferation. Guide 1 again had the most profound effect, reducing the expansion of the culture to about 60% of that seen in mock-transfected cells. Guide 3 reduced expansion to around 70% of mock-treated cultures, while guides 2 and 4 showed a modest effect (about 80%). Guide efficacy in this

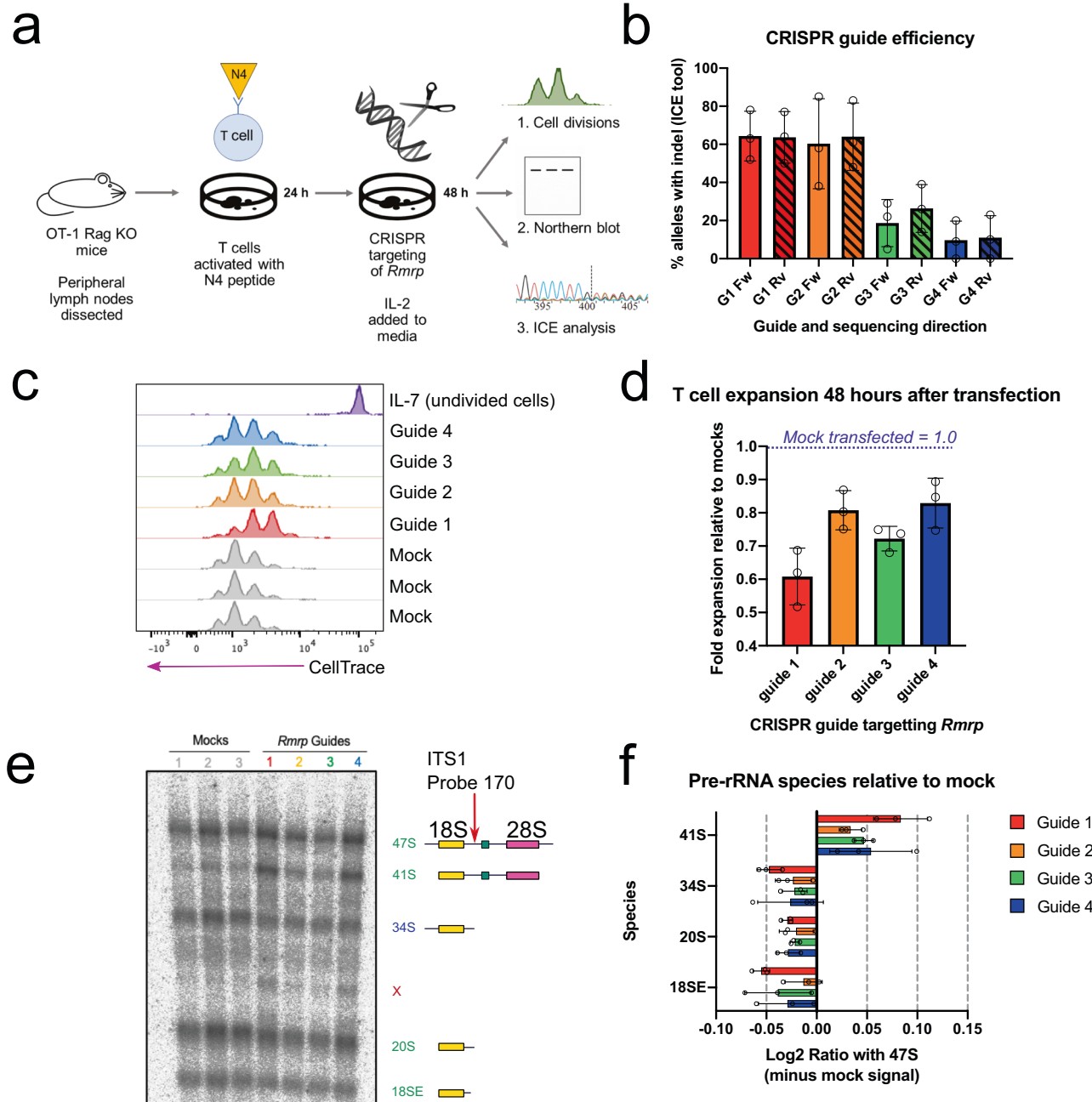

**Fig. 1 Disruption of *RMRP* impairs T cell proliferation and rRNA processing. a** Outline of experiment. Naïve TCR transgenic (OT-1) T cells were obtained from mouse peripheral lymph nodes and activated with N4 peptide. After 24 h, cells were electroporated with a CRISPR mix targeting *Rmrp*, or a mix without the guide (mock-transfected controls). After 48 h, cells were analyzed by flow cytometry for proliferation, northern blotting for pre-rRNA species, and ICE analysis to determine CRISPR guide efficiency. **b** CRISPR guide efficiency as determined by the Synthego Inference of CRISPR Edits (ICE) tool. A 669 bp region including *Rmrp* was amplified from DNA extracted from CRISPR-targeted cultures and sequenced with forward (Fw) and reverse (Rv) primers. The ICE tool deconvolutes Sanger sequencing traces to estimate the proportion of alleles in the population containing an indel. Graph summarizes data from three independent experiments. **c** T cell proliferation 48 h after CRISPR targeting of *Rmrp*, as measured with division tracker dye CellTrace Violet. Gated on live cells, as shown in Supplementary Fig. 2B. **d** Expansion of T cell cultures, calculated from data illustrated in panel **c**. Includes data from three independent experiments. In each replicate, expansion of CRISPR-targeted cultures was normalized to the average of mock-targeted controls in that experiment. **e** Pre-rRNA species detected by a probe in ITS1, in RNA from T cells 48 h after disruption of *Rmrp*. Cartoon on right shows species represented by each band. Pre-rRNA species indicated in green are present in both Pathway 1 and Pathway 2 (Supplementary Fig. 1). 41 S in brown is a Pathway 1 intermediate, and those in blue are Pathway 2 species. **f** Quantification of pre-rRNA species shown in panel **e**. X indicates a noncanonical precursor; analogous bands were seen in human cells with non-specifically disrupted *RMRP*[21]. Intensity values for each band were first normalized to the co-running 47 S/45 S band in that lane. For each rRNA species, the average intensity signal from the mock controls in that experiment was then subtracted. The resulting value was Log2 transformed. Panels **b**, **d**, and **f** show mean values from three independent experiments, with SD. Source data are provided as a Source Data file.

assay is predicted to depend on both guide cutting rates and the tolerance to small mutations of the target region in the RNA. These analyses were restricted to viable cells, as assessed by a dye for live/dead cells (Zombie Red), implying that non-lethal mutations in *RMRP* slow T cell division.

We next assessed whether these growth defects are associated with impaired rRNA processing. RNA was extracted from cultures 48 h after transfection and analyzed by northern hybridization, which allows each pre-rRNA species to be visualized and quantified (Fig. 1e, left). Pre-rRNA probe 170 hybridizes in ITS1 5′ to the site 2, thus detecting pre-rRNAs that have not undergone cleavage at site 2, potentially by RNase MRP (Fig. 1e, right cartoon, and Supplementary Fig. 1)[8,29]. For quantitation, pre-rRNAs levels were analyzed by the Ratio Analysis of Multiple Precursor (RAMP) method[30], in which the abundance of each pre-rRNA species is compared to the abundance of the 47 S primary transcript plus the initial processing product 45 S pre-rRNA. These are not resolved and are jointly designated 47 S (Fig. 1f). All cultures treated with *RMRP* guides showed elevated levels of 41 S pre-rRNA, the intermediate immediately downstream of 47 S, whereas later intermediates were depleted. These results indicate that removal of the 5′ external transcribed spacer (5′ ETS) was unaffected, but cleavage in ITS1 to separate the LSU and SSU precursors is delayed (Supplementary Fig. 1).

The pre-rRNA processing defect was most prominent after targeting with guide 1, the guide that had shown the greatest effect on proliferation and produced the largest proportion of mutated loci. However, the effect of other guides did not correlate well with their effect on proliferation. Guides 2 and 4 had similar effects on proliferation, but guide 4 caused more 41 S accumulation than guide 2.

Overall, these results support the model that disruption of *RMRP* impairs T cell proliferation by slowing pre-rRNA processing. However, disruption of different sites in the *RMRP* gene had different relative effects on cell division and pre-rRNA accumulation. We speculate that some guides preferentially cause lethal mutations in a smaller fraction of cells, while others induce milder defects in a larger population.

**Mutations associated with Cartilage Hair Hypoplasia impair pre-rRNA processing.** Fibroblasts from CHH patients have reduced growth rates and cell cycle defects[31]. We tested whether these problems are associated with impaired pre-rRNA processing, by measuring pre-rRNA species in patient-derived, CHH fibroblasts (Fig. 2a). Cells from healthy volunteers were used as controls. Being slow growing, these cells contained substantially less pre-rRNA than activated T cells, so obtaining high-quality RNA quantitation was challenging. Unlike *RMRP*-disrupted mouse T cells, the 41 S pre-rRNA in patient fibroblasts was not clearly increased relative to 47 S (Fig. 2b). 41 S pre-rRNA is generated in processing Pathway 1, by initial removal of both the 5′ ETS and 3′ ETS from 47 S, leaving the SSU and LSU rRNA precursors joined (Supplementary Fig. 1)[29]. In Pathway 2, initial cleavage at site 2 in ITS1 separates the 30 S (precursor for the SSU) and 32 S pre-rRNA (precursor for the LSU). Relative to 41 S, the 30 S pre-rRNA was significantly less abundant in CHH fibroblasts than controls. We propose that slowed site 2 cleavage, caused by reduced RNase MRP activity, leads to preferential utilization of Pathway 1 over Pathway 2 in CHH patient cells (Fig. 2b).

To overcome the challenges of working with pre-RNA from fibroblasts, we generated a cell line homozygous for the most common CHH-associated mutation, an A to G transition at position 71 on the current reference sequence (NBCI sequence

NR_003051.3; Supplementary Fig. 3a)[1]. Prior literature refers to this mutation as 70$^{AG}$ based on previous reference sequences, and we use 70$^{AG}$ for consistency. K562 cells, a suspension line derived from chronic myelogenous leukemia, were selected as the parental line[32].

The 70$^{AG}$ mutation was introduced by combining CRISPR/CAS cleavage with a single-stranded repair templates including the A => G mutation flanked by homology arms. Position 70$^{A}$ overlaps a PAM site compatible with CRISPR-Cas12a (previously Cpf1), allowing the 70$^{AG}$ mutation to be introduced without creating extraneous mutations in the ncRNA[33]. Four independent, homozygous CRISPR-derived clones were obtained (Supplementary Fig. 3b). Levels of the RMRP ncRNA were not significantly reduced in these cells compared to wild-type levels, when assessed by qPCR (Supplementary Fig. 3c). For unknown reasons, one clone (clone C) showed slightly reduced levels of the ncRNA RPPH1, which forms the related RNase P complex (Supplementary Fig. 3c)[12]. All tested 70$^{AG}$ clones grew on average more slowly than parental cells: over 48 h, mutant cell cultures expanded 25% less than wildtype controls (Supplementary Fig. 3d).

The cell lines gave good yields of high-quality RNA for pre-rRNA northern hybridization (Fig. 2c). RAMP analyses with an ITS1 probe (Probe 119; Fig. 2c left) showed that the 41 S:47 S ratio was consistently increased in 70$^{AG}$ mutants, with an average band intensity 0.08 (Log2) above wildtype. In contrast, the ratios of 30 S:47 S and 26 S:47 S were reduced, with an average change of −0.05 and −0.06, respectively (Fig. 2d). An ITS2 probe (Probe 123; Fig. 2c right), showed a visible reduction in 32 S relative to the 47 S band. Quantifying this showed that 32 S:47 S was decreased by an average of −0.15 fold. To further validate these results, we used qPCR to measure the ratio of pre-rRNA ITS amplicons to 5′ETS amplicons, both normalized to wildtype levels to correct for possible global perturbations of 47 S synthesis in these slower growing cells (Supplementary Fig. 3e, f). This showed that ITS-containing species were increased relative to the 5′ETS amplicons, consistent with an accumulation of 41 S. Overall, we conclude that more 47 S is processed in Pathway 1 (via 41 S) in 70$^{AG}$ mutants, with a decrease in Pathway 2 species (notably 30 S and 26 S), as found in CHH patient cells.

In *S.cerevisiae*, loss of MRP activity in yeast favors accumulation of the longer 5.8S$_L$ relative to the 6–8 nucleotide shorter form, 5.8S$_s$[14,15,17]. It was previously reported that fibroblasts from CHH patients have perturbations in the 5.8 S rRNA population, with an increased abundance of species with a 5′ extension into ITS1, as measured by qPCR[20]. We assessed the ratio of 5.8S$_S$ to 5.8S$_L$ in the 70$^{AG}$ K562 cells, using polyacrylamide gels for RNA separation to give nucleotide-level resolution (Fig. 2e). In northern hybridization, 5.8S$_S$ and 5.8S$_L$ were clearly resolved, with 5.8S$_S$ being more abundant. However, the 5.8S$_S$ to 5.8S$_L$ ratio was the same in parental and 70$^{AG}$ cells, at approximately 1.25 (Fig. 2f). It may be that the reported, extended 5.8 S rRNA qPCR products were generated by accumulated pre-rRNAs, particularly 41 S, which includes the ITS1-5.8 S region (Supplementary Fig. 1)[20].

**The 70$^{AG}$ mutation in *RMRP* reduces cytosolic ribosome abundance.** We next assessed if the processing delay in 70$^{AG}$ cells resulted in a lower abundance of mature rRNA species. This is difficult to quantify using northern blotting because rRNA comprises the majority of cellular RNA[34] as changes in rRNA abundance are masked when a set quantity of total RNA is loaded on the gel. Instead, we used flow cytometry to measure per-cell 18 S and 28 S RNA signals, using fluorescently-labeled oligonucleotide probes, a technique called FlowFISH (Flow-cytometry based fluorescently labelled in-situ hybridization)[6,35]. Probes with

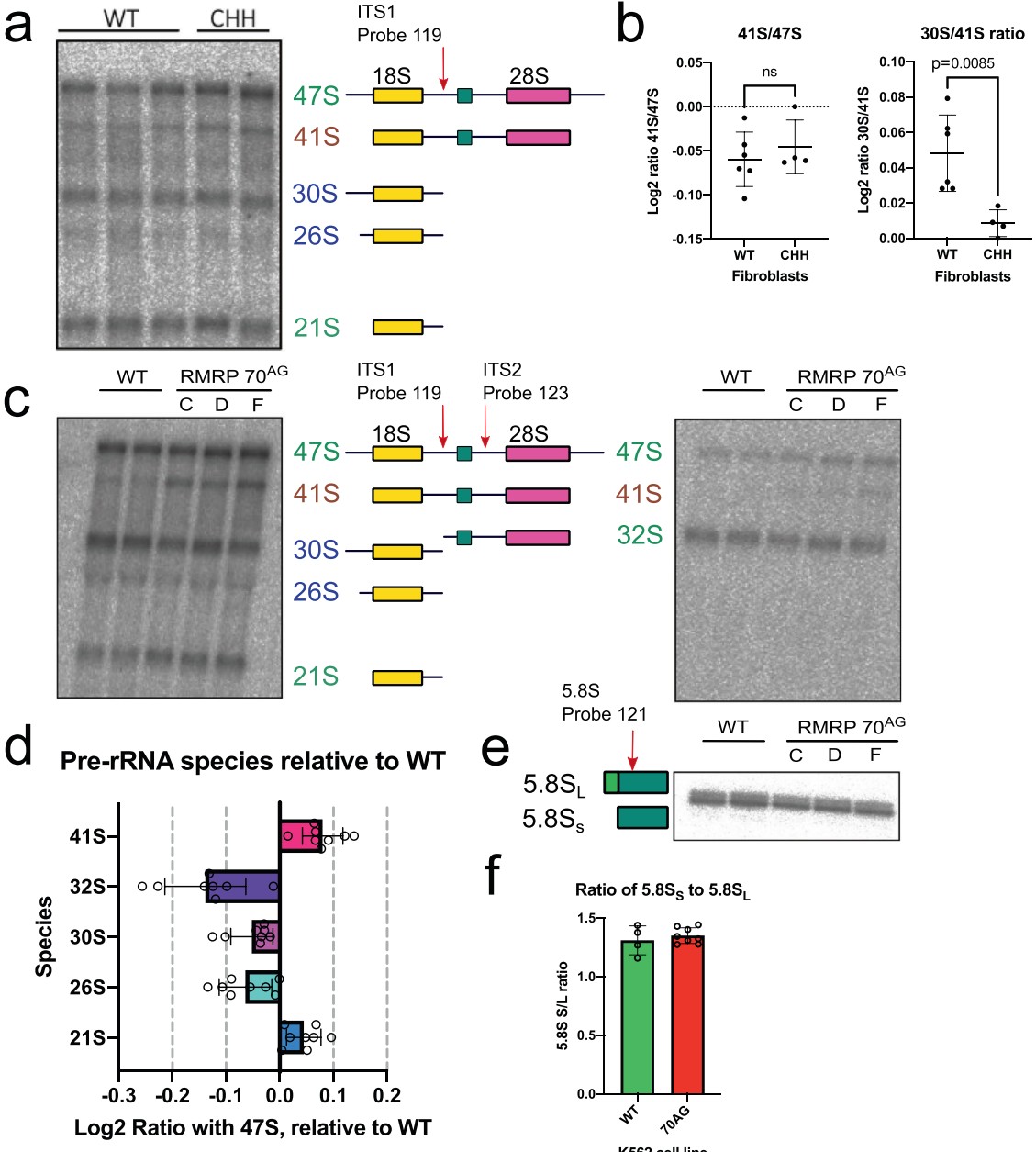

**Fig. 2 Mutations associated with Cartilage Hair Hypoplasia impair rRNA processing. a** Northern blot using ITS1 probe (probe 119) to detect pre-rRNA species in RNA extracted from CHH or healthy control (WT) fibroblasts. Pre-rRNA species indicated in green are present in both Pathway 1 and Pathway 2. 41 S in brown is a Pathway 1 intermediate, and those in blue are Pathway 2 species. **b** Quantification of relative abundance of 41 S to 47 S, and 30 S to 41 S, from northern blots as illustrated in **a**. Each dot represents an independent sample (total of 6 WT and 4 CHH samples; mean +/− SD). Includes samples processed in two independent northern blotting experiments. Indicated p value for 30/41 S derived from two-tailed t test (t = 3.463, df=8). **c** Northern blot using probes against ITS1 (left; probe 119) and ITS2 (right; probe 123), to detect pre-rRNA in parental K562 cells, or CRISPR-generated clones of K562 with a 70^AG mutation in *RMRP*. C, D, and F represent cell lines derived from independent CRISPR clones. **d** Quantification of pre-rRNA species from WT and 70^AG K562 cells. The intensity of bands for each indicated species were first normalized to the 47 S band in that lane. Values from WT cells were then subtracted from mutant values. Includes data from three independent experiments (mean +/− SD). **e** Northern blot with probe against 5.8 S rRNA, showing the short (5.8S_S) and long (5.8S_L) form of these species in indicated cell lines. **f** Quantification of ratio between 5.8S_S and 5.8S_L determined from northern blots illustrated in **e**. Includes data from three independent rounds of RNA extraction, run on two separate gels, including total of 4 wild type and 8 mutant samples (mean +/− SD). Source data are provided as a Source Data file.

scrambled nucleotide sequences were used as negative controls and gave low background (Fig. 3a). All 70^AG clones tested had significantly reduced rRNA signals compared to the parental lines (Fig. 3b), but there was some variability. Clone C showed the greatest reduction, with an rRNA signal at about 60% of parental cells. Clone D showed approximately 80% of wildtype signal, with clone F at 70%.

The ratio between 18 S and 28 S did not vary significantly between wild-type and mutant cells in the FlowFISH data. To support the conclusion that 18 S and 28 S rRNA were equally affected, RNA samples were run on a BioAnalyzer to quantify the relative fluorescence between mature rRNA species (Fig. 3c)[36]. As expected, the ratio of 28 S to 18 S fluorescence was about 2 in all samples tested (reflecting the greater length of 28 S rRNA)

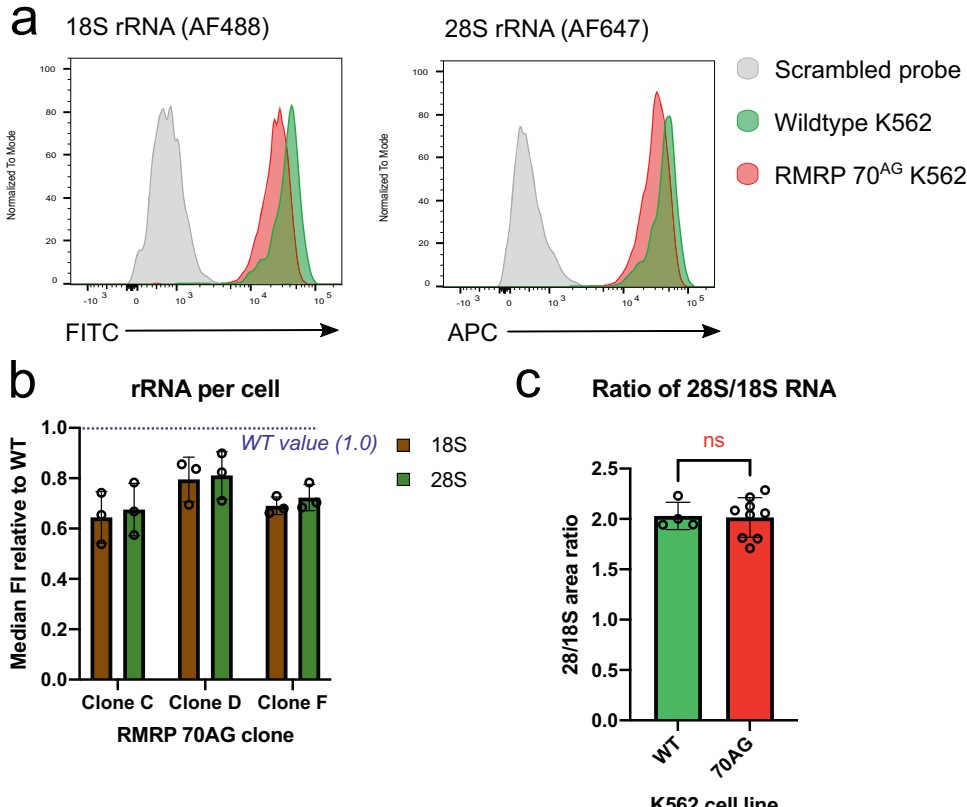

**Fig. 3 70$^{AG}$ mutation in *RMRP* causes a reduction in rRNA per cell. a** Example data from FlowFISH experiments, showing 18 S and 28 S rRNA detected by fluorescently-labeled antisense oligonucleotide probes, or signal from control probes with scrambled sequences. Gated on morphologically live cells. **b** Quantification of rRNA signals obtained in FlowFISH experiments. In each experiment, intensity values for each 70$^{AG}$ clone were normalized to the signal from wild-type cells. The gating strategy is shown in Supplementary Fig. 2c. The plot shows mean of data from three independent experiments, with SD. **c** Ratio of 28 S to 18 S rRNA in WT and 70$^{AG}$ cells, as determined by BioAnalyzer analysis. Mean and SD of data obtained in three independent experiments (total of 4 WT and 9 mutant samples). Nonsignificance concluded from two-tailed unpaired *t*-test ($p = 0.8963$, $t = 0.1334$, df=11). Source data are provided as a Source Data file.

with no significant differences between the mutants and parental cells.

As per-cell rRNA was reduced in 70$^{AG}$ mutants, we next tested whether they also had a reduction in mature ribosomes using <u>T</u>otal <u>R</u>NA-<u>A</u>ssociated <u>P</u>rotein <u>P</u>urification (TRAPP)[37]. This method quantifies the RNA-bound proteome, with proteins recovered in proportion to their interaction with RNA. To do this, in vivo RNA:protein complexes are stabilized by UV irradiation in living cells (Fig. 4a). Cells are then lysed in denaturing conditions, and RNA-associated proteins captured by binding the RNA portion of the complex to a silica column. Unbound proteins are washed away and the remaining silica-bound RNA-associated proteins digested with trypsin. Released peptides are eluted for analysis by mass spectrometry (LC-MS-MS). Wildtype and 70$^{AG}$ cells were directly compared by growing each in media with isotopically-labeled "heavy" or "light" amino acids, respectively, and mixing samples 1:1 by nucleic acid content. The output of the experiment is ratios of protein abundance between mutant and wildtype cells, called SILAC ratios (Stable Isotope Labelling with Amino acids in Cell culture)[37,38]. We anticipated that the 70$^{AG}$ mutants would show reduced relative recovery of ribosomal proteins (RPs) if mature ribosomes were less abundant in these cells.

About 1,300 SILAC ratios were recovered per mix (Fig. 4b). The mean ratio was 1 (log transformed to 0), confirming that there was no systematic bias between samples (Fig. 4b). As expected, mutant cells showed a small decrease in average abundance of cytoplasmic RPs, implying reduced interaction

between RPs and rRNA (Fig. 4c). Human cells have two separate populations of ribosomes, cytoplasmic and mitochondrial, which differ in both rRNA and protein composition[39]. The processing of mitochondrial ribosomes is independent of *RMRP*. Strikingly, recovery of cytoplasmic RPs from 70$^{AG}$ cells was significantly lower than recovery of mitochondrial RPs, relative to wild-type cells (RPMs; Fig. 4c). This difference was statistically significant ($p < 0.0001$). The same result was obtained in a repeat experiment where SILAC labelling was swapped such that mutant cells were "heavy" and wild-type cells "light", confirming that it was not a technical artifact caused by SILAC labelling.

There are two possible explanations for this result. First, 70$^{AG}$ cells might genuinely have more mitochondrial ribosomes. RNase MRP was originally identified as cleaving an RNA primer required for mouse mitochondrial DNA replication. The 70$^{AG}$ mutation could potentially increase the efficiency of this process and increase mitochondrial copy number[12]. Alternatively, the result could be caused by normalization to total RNA. In TRAPP, equal amounts of RNA from wildtype and mutant cells are mixed in order to purify RNA-bound proteins[37]. If mutant cells have less total RNA per cell, due to reduced rRNA abundance, more cell-equivalents will be included in the mix relative to the wild-type. This will cause an apparent reduction in ribosomal proteins in the TRAPP-purified proteome compared to other abundant RNA-interacting proteins, such as mitochondrial RPs.

Cytoplasmic RPs are also abundant relative to the total proteome so a similar effect would be expected for analyses of

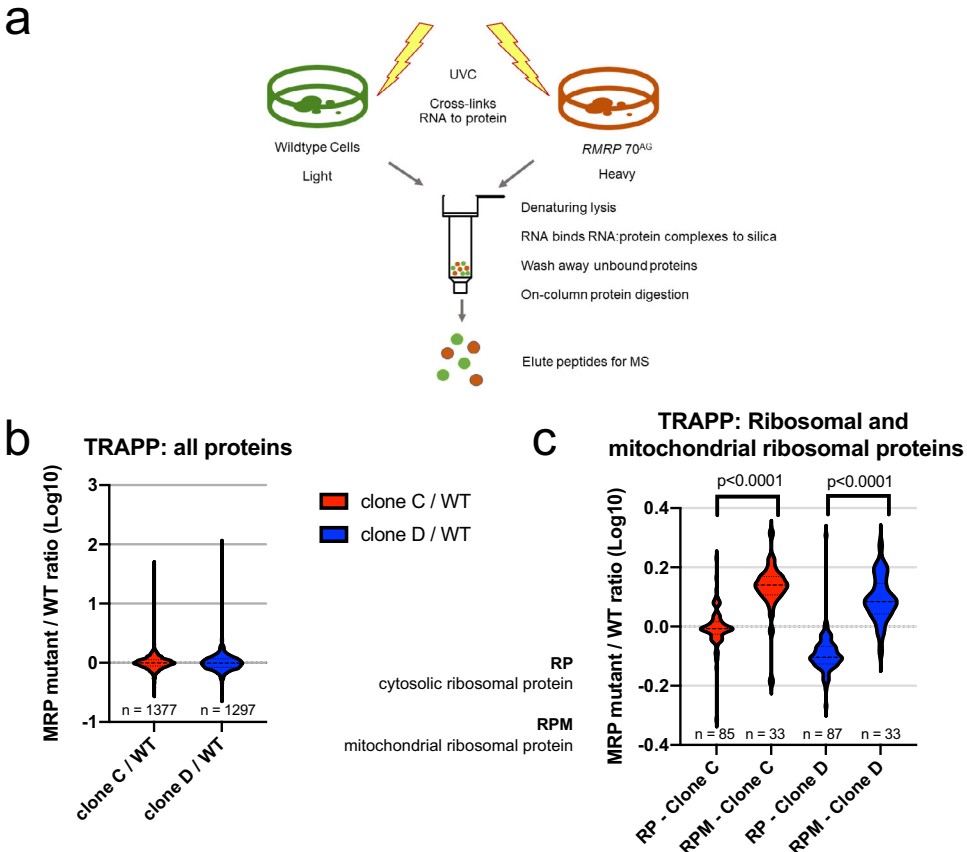

**Fig. 4 70^AG mutation in *RMRP* results in a reduced ratio of cytosolic to mitochondrial ribosomes. a** Overview of TRAPP technique. Wildtype and *RMRP* 70^AG cells were grown in heavy or light SILAC media, respectively, and UV crosslinked to stabilize RNA:protein interactions. Cells were lysed in denaturing conditions, and RNA:protein complexes bound to a silica column. Nonbound proteins were washed away, and remaining proteins digested on the column. Peptides were eluted for mass spectrometry. **b** Log-transformed SILAC ratios (70^AG mutant/wildtype) for all proteins quantified in TRAPP. **c** Log-transformed SILAC ratios for cytosolic ribosome proteins (RP) and mitochondrial ribosome proteins (RPM) obtained in TRAPP. Data shown in **b** and **c** are representative of two independent experiments, each including two SILAC mixes with different CRISPR clones. Violin plots depict distribution of SILAC ratios, with lines at median and quartiles. Indicated p values derived from two-tailed *t* tests: for clone C, $t = 9.458$, df = 116; for clone D, $t = 13.62$, df=118. Source data are provided as a Source Data file.

total protein. Indeed, the same trend was seen in SILAC total proteome data (Supplementary Fig. 4a, b).

**70^AG mutation in *RMRP* reduces the abundance of intact RNase MRP complexes.** RNase MRP is a ribonucleoprotein complex comprised of the RMRP ncRNA and about 10 proteins[12]. All of these are reported to be shared with an evolutionarily related ribonucleoprotein complex, RNase P, which cleaves the 5′ leader from pre-tRNAs. RNase P includes the ncRNA RPPH1 in place of RMRP[12] and has a similar abundance to MRP[40,41]. Mutations in the shared POP1 protein cause diseases that overlap with skeletal phenotypes of *RMRP* mutations[42–44]. In recent cryo-EM structures of yeast RNase MRP[45,46], Pop1 was shown to be the main structural protein in the complex. Its C-terminus interacts with the C-domain of the yeast RMRP ncRNA (called NME1), while the N terminal is in contact with the S-domain. No high-resolution structure of human RNase MRP has yet been reported.

In TRAPP data, POP1 and another RNase MRP/P complex protein, RPP38, showed consistently reduced RNA-association in 70^AG mutants compared to controls (Fig. 5a). In analyses of total protein, POP1 was slightly less abundant in mutants than parental cells, but this was not statistically significant (SD crossing 1; Fig. 5b). The results raised the question of which

RNAs are most associated with POP1 in vivo, as these interactions must be reduced in 70^AG cells to cause the reduced recovery of POP1 in TRAPP. To address this, we used the technique of crosslinking and analysis of cDNAs (CRAC; shown schematically in Supplementary Fig. 5) to map RNAs associated with individual MRP/P complex proteins[47]. In addition to POP1, we performed this analysis on POP4, as the yeast structures suggested this protein to be in close proximity to the known pre-rRNA substrate[46]. A CRISPR-based approach was used to insert a FLAG-HIS₇ tag onto the N or C termini of these proteins in K562 cells. Clones with homozygous tags were selected for analysis and RNA:protein complexes were stabilized by UV crosslinking in vivo. The bait protein was then tandem purified in stringent, denaturing conditions, and co-purifying RNAs sequenced[47].

Notably, RMRP comprised the majority of RNAs recovered with POP1, making up 77–82% of sequencing reads (Fig. 5c). RPPH1 was significantly less recovered with POP1, accounting for 17–23% of reads. As the abundance of RMRP was not significantly reduced in 70^AG mutants (Supplementary Fig. 3c), this result indicates that the reduced total RNA interactions of POP1 in 70^AG cells likely reflects impaired POP1:RMRP interactions. POP4 CRAC recovered the opposite ratio from POP1, with 79 – 86% of reads representing RPPH1 (Fig. 5C). Sites of RNA:protein crosslinks in CRAC are often revealed by single-base deletions in the sequencing data[47]. Mapping such crosslinks

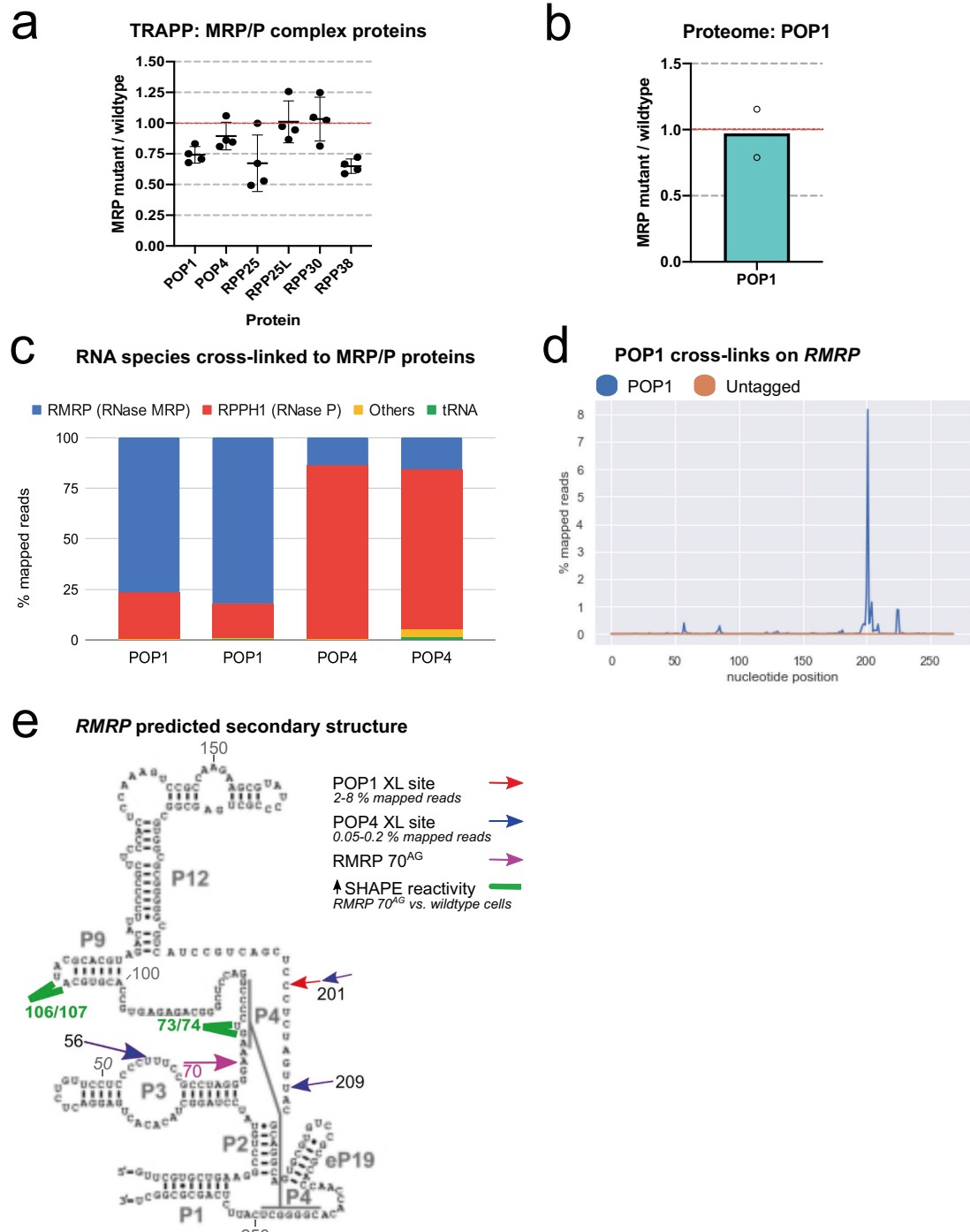

**Fig. 5 70^AG mutation in *RMRP* reduces the abundance of intact RNase MRP complexes. a** SILAC ratios (70^AG/wildtype cells) obtained for MRP/P complex proteins in TRAPP (mean and SD). Each dot represents data from an independent SILAC mix (total of 4 mixes, processed in 2 independent experiments). **b** SILAC ratios (70^AG/wildtype cells) obtained for POP1 protein in proteome. Includes data from two SILAC mixes, showing mean value. **c** RNA species recovered from MRP/P complex proteins in CRAC. Graphs shows relative proportion of mapped reads representing each indicated RNA or RNA biotype, in two independent experiments. **d** Pileup of reads containing single base deletions (indicating RNA:protein crosslink site) in RMRP ncRNA, in POP1 or negative control CRAC. Representative of results from two independent experiments. **e** Possible secondary structure of RMRP, indicating sites of crosslinking with two RNase MRP complex proteins (POP1 and POP4). Nucleotides with reproducibly increased SHAPE reactivity scores in *RMRP* 70^AG cells are indicated. The line designated P4 indicates a predicted interaction that generates a conserved pseudoknot structure. Source data are provided as a Source Data file (panels **a**, **b** and **c**), and as processed data files (panel **d**).

across RMRP showed a clear, single site of interaction with POP1, centered on nucleotide 201 (Fig. 5d), whereas crosslinks on RPPH1 were more diffuse with several smaller peaks (Supplementary Fig. 6a, b). Conversely, POP4 CRAC showed two clear peaks in RPPH1 (at positions 122 and 186), but multiple smaller peaks for RMRP (Supplementary Fig. 6c, d). There was some overlap in the recovered crosslinking sites for POP1 and POP4. This has previously been seen for other RNPs (see for example[48]). Our interpretation is that in vivo RNPs can show flexibility, perhaps on different substrates or at different steps during their function.

To determine whether the preference for POP1 crosslinking to the RNA component of MRP over P was conserved in yeast, we tested yeast Pop1 in CRAC. This showed that, indeed, the bias was even more pronounced: 89–94% of reads were from the RNase MRP RNA (NME1 in yeast) and 3–7% from RNase P (RPR1; Supplementary Fig. 7a). However, unlike human POP1, the yeast protein showed three distinct crosslink sites across the MRP RNA (Supplementary Fig. 7b), suggesting that the structures of human and yeast RNase MRP complexes may not be identical in vivo.

POP1 apparently shows a conserved, preferential association with RMRP ncRNA in vivo. Levels of POP1 and RMRP are maintained in 70[AG] cells, so the reduced POP1 recovery in TRAPP indicates that the mutation likely destabilizes the POP1:RMRP interaction. Potentially this could be associated with altered folding of the RMRP RNA in vivo. This hypothesis is supported by the finding from yeast structural data that, although human disease-causing mutations in RMRP cluster near the catalytic center of RMRP, these sites do not appear to bind the known the substrate of yeast MRP (the A3 fragment of ITS1 in pre-rRNA), implying that they may act by impairing RNA folding or complex stability.

We assessed this possibility using selective 2′-hydroxyl acylation analyzed by primer extension and mutational profiling (SHAPE-MaP)[49–51]. In this method, an electrophilic chemical probe (1M7 in this case) is added to cultured cells and modifies RNA nucleotides in proportion to their flexibility. Sites of modification are detected as single-base mutations in sequencing data after gene-specific reverse transcription and cDNA sequencing. Overall, the obtained RMRP SHAPE reactivity profiles were similar between wildtype and 70[AG] cells (Supplementary Fig. 8), indicating no large-scale refolding in the mutants. Visual inspection revealed few changes, however, the deltaSHAPE tool[50] performs statistical comparisons based on average reactivity at each nucleotide (Supplementary Fig. 9). This identified two areas showed small, but apparently reproducible, increases in flexibility in 70[AG] cells (indicated in green in Fig. 5e and boxed in Supplementary Fig. 9), one just downstream of the mutation (nucleotides 73 and 74) and one further 3′ (106 and 107).

We conclude that the disease-associated mutation likely results in subtle structural destabilization of the RNase MRP complex.

## Discussion

Ribosomopathies are a diverse group of human disorders in which ribosome production or function is defective[23–25]. Clinical features of these disorders are variable; bone marrow dysfunction leading to anemia or cytopenia is common, as are skeletal anomalies and an increased risk of cancer. Like CHH, many ribosomal disorders have surprisingly tissue-specific phenotypes, despite ribosomes being present in almost all cells[26,27]. Two broad hypotheses have been proposed to explain this. First, the phenotype may reflect decreased ribosome number or function in specific cell types that crucially depend on poorly translated mRNAs, vulnerable to reduced translation. In Diamond Blackfan anemia (DBA), in which mutations in ribosomal proteins cause defective erythropoiesis, ribosome numbers are reduced without altered composition[52]. This results in reduced translation of transcripts with unstructured 5′ untranslated regions (UTRs), including a specific reduction in the translation of GATA2, a key hematopoietic transcription factor. Alternatively, and not mutually exclusively, some cell types may rely on specialized ribosomes, the assembly of which may be affected by particular mutations[25].

Here, we present evidence that CHH is a ribosomopathy caused by a defect in pre-rRNA processing, Previous work reported increased levels of ITS1-containing rRNA precursors in CHH patient cells[20,53], but downstream effects on ribosome abundance have not previously been demonstrated. In this study, patient fibroblasts and a cell line with a disease-linked mutation (70[AG] in *RMRP*) showed delayed rRNA processing in a pattern consistent with decreased cleavage at the presumed RNase MRP target (site 2 in ITS1). These cells also have reduced rRNA per cell, and reduced intact cytosolic ribosomes relative to the mitochondrial ribosome pool. Moreover, the 70[AG] mutation reduces the abundance of intact RNase MRP complexes, probably by destabilizing the interaction between RMRP and POP1. The overall effects on ribosome synthesis are modest, but this is expected from a disease-related mutation in an essential gene, since carriers must develop almost normally in order to be classed as patients.

A high-resolution structure of the human RNase MRP complex is not currently available, and the POP1/Pop1 crosslinking sites suggest that there may be appreciable differences from the yeast structure. Based on evolutionary comparisons[41,54], the predicted secondary structure of RMRP is shown in Fig, 5e. The line designated P4 (Fig. 5e) indicates a predicted interaction that generates a conserved pseudoknot structure. This will fold the core of the RNA into a compact 3D structure postulated to play a role in catalyzing substrate RNA cleavage. Within this structure, all indicated RNA sites will likely be in close proximity. The large POP1 protein (115 kDa) crosslinked at a single site on RMRP (Fig. 5e). In contrast, the smaller POP4 protein (25 kDa) crosslinked at multiple sites. Notably, one of these POP4 binding sites overlapped with one of the POP1 crosslinking sites, suggesting conformational changes within the complex, perhaps during different steps during pre-ribosome association and function. Chemical probing of structural flexibility in vivo using SHAPE-MaP[49–51] did not reveal substantial changes in the mutant, although modest changes in the structure are indicated (green in Fig. 5e) at sites in the vicinity of the mutation. We speculate that changes in the overall structure of the MRP core impact negatively on the stability of protein interactions and catalytic activity in vivo.

These results advance our understanding of CHH, but also raise some intriguing questions. Most notably, why are the phenotypes of DBA and CHH different, if reduced ribosome number is a common pathological mechanism? Some aspects of the two disorders do overlap; for example, some CHH patients have bone marrow dysfunction similar to that seen in DBA[55]. However, other aspects are unique, notably the T cell dysfunction which is life-limiting in CHH but not a feature of DBA[1].

In conclusion, the results in this study point to CHH being a disorder of ribosome synthesis and suggest experimental approaches to further explore this complex disease.

## Methods

**Human cell culture**. K562 cells were obtained from ATCC (cat. CCL-243), and grown in RPMI 1640 Medium with GlutaMAX Supplement (Gibco; cat. 61870036), further supplemented with 1x final concentration Antibiotic-Antimycotic (Gibco; cat. 15240096) and 10% fetal calf serum (Sigma; cat. F2442). Cells were grown to a density of 0.5–1 × 10^6 cells/mL, then diluted or used for experiments.

Patient and control fibroblasts were obtained from Prof. Sophie Hambleton (Newcastle University). They were grown in DMEM with GlutaMax (Gibco; cat. 10566016) supplemented with 10% fetal calf serum. Cells were grown to a confluency of 70–90% then split or used for experiments. To split, cells were washed once with sterile PBS, then incubated with 0.25% Trypsin-EDTA (Gibco; cat. 25200056; 0.2x volume of culture media removed) until detached. Trypsin was inactivated with 5 volumes of media, and cells pelleted at 100 RCF for 5 minutes before resuspension in culture media. All mammalian cell cultures were maintained at 37 °C with 5% $CO_2$.

**Flow cytometry**. Samples were prepared as described for individual methods, below. Samples were acquired on a BD LSRFortessa flow cytometer, using BD FACSDiva software, version 8.0.1. Initial data analysis was done with FlowJo version 10.6, with subsequent analysis as indicated in figure legends.

**CRISPR targeting of *Rmrp* in primary mouse T cells**. Rag1 KO, C57BL6/J mice were bred at the University of Edinburgh. All experimental procedures were approved by a current project license under the authority of the Animals (Scientific Procedures) Act 1986, and additionally followed the University of Edinburgh's ethical guidance as overseen by its AWERB committee.

Peripheral lymph nodes were dissected from Rag1 knockout mice homozygous for the OT1 allele. Lymph nodes were massaged through a 70 μM cell strainer. Cells were washed once with IMDM (Gibco; cat. 12440053) supplemented with 2 mM L-glutamine, counted, and resuspended at $5 \times 10^6$ cells/mL in PBS supplemented with 2.5 μM CellTrace Violet (Invitrogen; cat. C34557). After 20 min at 37 °C, cells were washed with complete T Cell media (TCM; IMDM supplemented with 10% FCS, 2 mM L-glutamine, 100 U/mL penicillin, 100 U/mL streptomycin, and 50 μM 2-mercaptoethanol), counted, and resuspended at 250,000 cells/mL in TCM supplemented with 10 nM N4 peptide (peptide sequence SIINFEKL).

After 24 h of stimulation, cells were counted, pelleted, and resuspended for transfection at 1.5 million cells per transfection in 80 μL Neon Transfection System Buffer T (Invitrogen; cat. MPK10096). CRISPR guide RNAs were purchased from IDT (Supplementary Table 1), and resuspended at 100 μM in Nuclease-Free Duplex Buffer (IDT; cat. 11-01-03-01). For each transfection, 2.5 μL of guide RNA was mixed with 2.5 μL of tracrRNA and 20 μL of Nuclease-Free Duplex Buffer. The mix was heated at 95 °C for 5 min and then allowed to cool. Once at room temperature, 23 μL of Buffer T and 2 μL of TrueCut Cas9 Protein v2 (5 mg/mL stock; Invitrogen; cat A36496) was added, and the combined mix heated to 37 °C for 10 min before electroporation using a Neon Transfection System with $3 \times 1600$ V pulses of width 10 ms. Electroporated cells were then cultured for 48 h at either 250,000 (for flow cytometry) or $1 \times 10^6$ cells per ml (for northern blotting) in TCM supplemented with recombinant IL-2 (20 ng/mL). For flow cytometry, cells were first washed with PBS before live/dead staining with Zombie Red Fixable Viability Kit (1:100 in PBS for 20 mins at room temperature; BioLegend; cat. 423109). Cells were then washed twice with FACS buffer before sample acquisition.

For proliferation experiments, fold expansion of culture was calculated by summing all the cells in this gate, and dividing this by the calculated number of initial cells. Initial cell number was calculated by first calculating the number of cells in each division peak, and dividing this by $2^x$, where $X$ is the number of divisions undergone by cells in that peak. Then, this value for all peaks was summed, giving the final initial cell value.

**ICE analysis of CRISPR-targeted mouse T cells**. 250,000 T cells were used as the input for DNA extraction, PCR and sequencing, using primers NR87 (5′-CCCACCTA GCGTTCCTACAT-3′) and NR88 (5′-AGAAATAAAAGTGGCCGGGC-3′). Sanger sequencing traces were analysed using the Synthego Inference of CRISPR Edits (ICE) tool web interface[28].

**Northern blotting**

*RNA extraction*. Mammalian cells were lysed in Trizol (Invitrogen; cat. 15596026). 200 μL of chloroform was added per 1 mL of Trizol used for cell lysis, and the mixture incubated at room temperature for 5 min. Phase separation, RNA precipitation, and washes were done according to manufacturer's protocol. After the wash step, the RNA pellet was resuspended in 100% formamide for northern blotting, or water for qPCR. To ensure the RNA was fully dissolved, the mixture was left on ice for 20 min followed by heating at 65 °C for 10 min. A 1:10 dilution of RNA mix was quantified on a spectrophotometer.

*Acrylamide gel for short RNA species*. Loading buffer was made by supplementing 95% formamide with 20 mM EDTA, made up with MQ. One crumb of Xylene cyanole and bromophenol blue was added per 5 mL of this buffer. RNA samples were mixed 1:1 with loading buffer and heated to 65 °C for 5 min, and then incubated on ice before loading.

Gel mix containing 8.3 M urea was prepared by mixing 50 mL of 40% acrylamide gel solution (bis-acrylamide ratio 19:1; Severn Biotech; cat. 20–2400), 125 g of urea, and 25 mL of 10x TBE buffer, and diluting to 250 mL with MQ. Before pouring, 300 μL of 10% ammonium persulfate and 30 μL of TEMED (Sigma; cat. GE17-1312-01) were added per 50 mL of gel mix.

Gels were run for 1600 V hours (for example, 80 V for 20 h), and samples transferred onto BrightStar-Plus Positively Charged Nylon Membrane (Invitrogen; cat. AM10100) in 0.5x TBE for 3 h at 40 V. The membrane was then crosslinked at 254 nM in a Statralinker cross-linking device (Stratagene).

*Agarose gel for long RNA species*. 50x TRI/TRI buffer was prepared by mixing 10 mL of triethanolamine and 13.5 g tricine, and bringing the volume to 50 mL with MQ. Agarose gel mix was prepared by combining 285 mL of MQ, 3 g of agarose and 6 mL of 50x TRI/TRI. Agarose was melted in a microwave and allowed to cool to just above room temperature, at which point 10.5 mL of 36% formaldehyde was added and the gel poured.

Gel loading dye was prepared by mixing 84 μL of 50x TRI/TRI, 4 μL of 0.5 M EDTA, 80 μL of 1% Bromophenol Blue and 1.84 mL of MQ. Pre-mix was then prepared by mixing loading dye, 36% formaldehyde, and 1 mg/mL ethidium bromide in the ratio 14:1:1. RNA samples were mixed with an equal volume of pre-mix, and heated at 70 °C for 10 min, followed by cooling on ice for 5 min before loading. The gel was run at 140 V for 4.5 h. After visualization, RNA was transferred to BrightStar-Plus Positively Charged Nylon Membrane by overnight downward capillary transfer in 10x SSC. Membranes were then crosslinked, as above.

*Radioactive olionucleotide probing of northern membranes*. 100x Denhardt's solution was prepared by dissolving 1 g of BSA fraction V, 1 g of Ficoll 400, and 1 g of PVP in 50 mL of MQ. Hybridization solution was prepared by combining 25 mL of 20x SSC, 5 mL of 10% SDS, and 5 mL of 100x Denhardt's solution, and bringing to 100 mL with MQ. Oligo labelling reaction mix was prepared by mixing 1 μL of 10 mM oligo with 13 μL of MQ, 2 μL of 10x T4 PNK Reaction Buffer (NEB; cat. B0201S), 1 μL of 100 mM DTT, 1 μL of T4 PNK (NEB; cat. M0201L) and 3 μL of $^{32}$P-$\gamma$ATP (10 μCi/ μL; Hartmann Analytic). The labelling reaction was performed at 37 °C for 1 hour. In the meantime, the membrane was pre-hybridized in hybridization solution at 50 °C with shaking. Following labelling, unincorporated ATP was removed from the reaction mix using a mini-Quick Spin Oligo Column (Roche; cat. 1814397001), following provided instructions. The recovered reaction mix was added to the membrane in 100 mL of hybridization solution for overnight hybridization.

The following morning, the membrane was washed four times with hybridization wash solution (2× SSC, 0.1% SDS). Two washes were done at room temperature, followed by a wash at 50 °C before a final room temperature wash. Membranes were then exposed to a phosphoimager screen. Before re-hybridization with a different probe, membranes were stripped twice by incubating for 10 min with boiling stripping solution (0.1× SSC, 0.1% SDS). Sequences of oligonucleotide probes used are shown in Supplementary Table 2. Acquired images were analyzed with ImageJ version 1.51.

**Generation of stable CRISPR-edited human cell lines**. For all human CRISPR experiments, guide RNAs, repair templates, and check primers were ordered from IDT from the sequences in Supplementary Table 3.

**Protein tagging in human cell lines**. For protein tagging experiments, single-stranded templates were designed including the tag with homology arms. For POP1, a 2xFLAG-6xHIS tag was inserted at the N terminus. For POP4, a C terminal 8xHIS 1xFLAG tag was used, with the HIS and FLAG moieties separated by a 4x Ala linker.

To introduce the tags, 0.6 μL of 200 μM guide RNA was mixed with 0.6 μL of 200 μM tracrRNA (Alt-R CRISPR-Cas9 tracrRNA (IDT; cat. 1072532), heated to 95 °C for 5 min then allowed to cool. Then, 2.1 μL of PBS and 1.7 μL of 61 μM Cas9 enzyme was added (Alt-R S.p. Cas9 Nuclease V3; IDT; cat. 1081058) and the mixture incubated at room temperature for 20 min. Meanwhile, $1 \times 10^6$ cells per transfection were pelleted, washed with PBS, and resuspended in 100 μL of Nucleofector Solution from the Cell Line Nucleofector Kit V (Lonza; cat; VVCA-1003).

The transfection mix was assembled with 5 μL of Cas9:gRNA mix (made above), 1 μL of 100 μM electroporation enhancer (IDT; cat.1075915), and 3 μL of 10 μM ssODN. Cells and transfection mix were combined and electroporated with a Lonza Nucleofector 2b Device, using supplied settings for K562 cells. Cells were then gently transferred to 1.5 mL of K562 media supplemented with 25 μM HDR Enhancer (IDT; cat. 1081072). After 48 h, cells were sorted by FACS to 1 cell / well in a 96 well plate. Clones were screened by PCR and sequencing after about 2 weeks, using check primers shown in Supplementary Table 3.

**CRISPR-Cpf1 mediated knock-in of RMRP 70$^{AG}$ mutation**. A single-stranded repair template was designed including the mutation with homology arms. 1.6 μL of 100 μM Cpf1 crRNA was mixed with 1.4 μL of PBS and 2 μL of Cpf1 Nuclease 2NLS (IDT; discontinued) and incubated for 20 min at RT. Thereafter the procedure was the same as for CRISPR-Cas9 mediated protein tagging, except the electroporation enhancer was Cpf1-specific (IDT; cat. 1076300).

**HIS-TEV-Protein A tagging of yeast Pop1**. Yeast stains in this study were derived from *Saccharomyces cerevisiae* strain BY4741[56]. HIS-TEV-Protein A (HTP) inserts with homology arms were amplified from plasmid pBS1539 using primers listed in Supplementary Table 4. The transformation protocol described here is adapted from Gietz and Woods[57]. An overnight yeast culture was diluted to $0.5 \times 10^7$ cells/mL (5 mL per transformation) and grown for 2 divisions. Cells were pelleted by centrifuging at 2000 g for 2 min at room temperature, washed with MQ, and resuspended in transformation mix composed of 240 µL PEG 3350 50% w/v, 34 µL PCR product, 50 µL pre-boiled salmon sperm DNA and 36 µL 1 M LiAc.

The mix was vortexed and incubated for 40 min at 42 °C. Cells were then pelleted at 14,000 g for 30 s at room temperature, resuspended in 80 µL MQ, and spread on a YPD plate. After two days, colonies were streaked onto selective medium (-URA).

**qPCR**. RNA was extracted from $1 \times 10^6$ cells, and 5 µg of RNA diluted to 10 µL in MQ. Reverse primer mix was made by combining 15 µL of all the reverse primers listed in Supplementary Table 5, each at 100 µM. 2.5 µL of this mix was added to each 10 µL RNA sample, and the combined mix heated to 72 °C for 5 min, after which the mix was split into two: one portion for reverse transcription and the other for a no-reverse transcription control.

Reverse transcription mix was made by combining 1.75 µL of MQ, 0.75 µL of 10 mM dNTP solution, 0.1 µL RNasin Ribonuclease Inhibitor (Promega; cat. N211A), 0.4 µL of Superscript IV Reverse Transcriptase (Invitrogen; cat 18090010) and 2 µL of provided reaction buffer. Mix and RNA samples were combined, incubated at 55 °C for 1 hour, and diluted 1:200 for use.

qPCR reaction mix was made by combining 2 µL of cDNA or no-reverse transcription control, 0.1 µL of MQ, 0.2 each of forward and reverse primers (at 10 µM; Supplementary Table 5) and 2.5 µL 2x SYBR Green PCR Master Mix (Applied Biosystems; cat. 4344463). Each reaction was set up in triplicate. cDNA was amplified in a LightCycler 480 (Roche), using the following cycle: initial 5 min at 95 °C, then 40 cycles of 10 s at 94 °C, 10 s at 60 °C and 15 s at 72 °C.

$C_T$ value for each amplification curve was determined by the LightCycler software (version 1.5), and averaged for technical triplicates. Matched $C_T$ for the house-keeping gene was subtracted to give $\Delta C_T$. Fold change of mutant (Mut) cells compared to wildtype (Wt) cells was calculated with the formula: $2^{-(\Delta C_T \, Mut - \Delta C_T \, Wt)}$. For some experiments, further analyses were undertaken, as indicated in Figure legends.

**FlowFISH**. The FlowFISH method[35] was adapted. Probe sequences shown in Supplementary Table 6 are published[58]. Probes conjugated to fluorophores were ordered from IDT.

*Fixation and permeabilization.* $0.5 \times 10^6$ K562 cells were used. Cells were pelleted at 500 g for 5 min at room temperature, and washed once with 0.5 mL of PBS. After pelleting, cells were resuspended in 0.5 mL of PBS, and 0.5 mL of 8% paraformaldehyde added. Cells were left to fix for 30 min at room temperature, after which they were washed twice with 1 mL of PBS and resuspended in 0.5 mL PBS. 0.5 mL of 70% ethanol was then added dropwise, and cells pelleted again before resuspension in 1 mL of 70% ethanol and permeabilization overnight at 4 °C.

*Rehydration and probing.* FlowFISH wash buffer (FFWB) was prepared by supplementing 2x SSC with 10% formamide and 0.25 mg/mL Bovine Serum Albumin fraction V (BSA; Sigma; cat. 05482).

FlowFISH solution A (FFSA) was prepared by combining, per sample: 5 µL of formamide; 2.5 µL of 2x SSC; 2.5 µL of 10 mg/mL *E. coli* tRNA (Sigma; cat. R1753); 2.5 µL of FISH probes diluted to 50 ng/µL; and 8.75 µL of MQ. FFSA was then heated to 95 °C and allowed to cool. Meanwhile, FlowFISH solution B (FFSB) was prepared by combining, per sample: 25 µL of 20% dextran sulphate dissolved in 4x SSC; 1.25 µL of 10 mg / mL BSA; and 40 units of RNasin Ribonuclease Inhibitor (Promega; cat. N211A). Once FFSA was cool, FFSA and FFSB were combined 1:1 to create the staining mix.

For staining, cells in ethanol were pelleted at 1000 g for 5 min, resuspended in 1 mL of FFWB and left to rehydrate at room temperature for 5 min before again being pelleted. Cells were then resuspended in staining mix for 3 h at 37 °C, before 2 washes with 1 mL FFWB, and 2 washes with 1 mL FACS buffer.

**In-cell SHAPE-MaP**. The SHAPE-MaP protocol described here is adapted from work published by the Weeks lab[49–51,59]. Wildtype or RMRP 70[AG] K562 cells were grown to log phase, then washed with PBS and $0.8 \times 10^6$ cells resuspended in 900 µL media. The SHAPE reagent 1M7 (Sigma; cat. 908401) was added to a final concentration of 10 mM by adding 100 µl of 100 mM reagent resuspended in DMSO. For unmodified samples, the same volume of DMSO was added. The in-cell acylation reaction was left to proceed for 15 min at 37 °C. Cells were then pelleted, washed with PBS, and lysed in Trizol (Invitrogen; cat. 15596026). RNA was extracted using the manufacturer's phase separation protocol, and 500 ng RNA diluted in 500 µL MQ. Reverse transcription mix was made by adding 4 µL of 2.5 mM dNTP and 1 µL of RMRP RT primer (5′-ACAGCCGCGCTGAGA-3′) at 2 uM. The mixture was heated to 65 °C for 5 min, then cooled on ice.

Next, the mixture was supplemented with 4 µL of 5x FirstStrand buffer (provided with SuperScript II; Invitrogen; cat. 18064014), 4 µL of freshly prepared 30 mM MnCl$_2$, 2 µL of 100 mM DTT, and 1 µL of RNasin Ribonuclease Inhibitor (Promega; cat. N2511), and incubated at 23 °C for 2 min. 1 µL of the SuperScript II enzyme was then added. Reverse transcription was then done in a thermocycler using the following programme: 25 °C for 10 min; 42 °C for 90 min; then 10 cycles of 50 °C for 2 min and 42 °C for 2 min. The enzyme was deactivated at 72 °C for 10 min. Buffer components and the RT primer were removed by purifying the cDNA mixture using a mini-Quick Spin Oligo Column (Roche; cat. 1814397001), eluting with 15 µL MQ.

To prepare a sequencing library, two sequential PCRs were performed. First, 5 µL of the purified cDNA was amplified using the primers NRs2-MRP-PCR1-FW and NRs3-MRP-PCR1-RW (Supplementary Table 7, adapted from[60]), in a 50 µL reaction mix containing 10 µL of 5x Q5 buffer (provided with Q5 polymerase kit; NEB; cat. M0491L), 4 µL of 2.5 mM dNTP, 2.5 µL of each primer at 10 µM, and 0.5 µL of the Q5 polymerase. The reaction was performed using the following program: initial denaturation at 98 °C for 10 s, then 30 cycles of 98 °C for 10 s, annealing at 60 °C for 30 s and extension at 72 °C for 20 s. The final extension was at 72 °C for 2 min. The reaction was purified using AMPure XPbeads, as described in the Library Preparation subsection of "CRAC protocol for human cell samples", below.

1 ng of this purified product was used as the input template for the second PCR, assembled essentially as before. Primers for this reaction were NRs4-PCR2-RW, and one of the barcoded universal SHAPE forward primers from Supplementary Table 7. The annealing temperature for this second reaction was 66 °C. The final libraries were gel purified, diluted to 2 ng / µL and pooled. Libraries were sequenced on an Illumina NextSeq 550 instrument using a mid-output flow cell (2×150 reads).

SHAPE reactivity profiles and comparisons were generated using the ShapeMapper 2 and deltaSHAPE scripts, using default settings and aligning to *RMRP*. Mutant samples were aligned to a modified *RMRP* reference sequence containing the 70[AG] mutation, to prevent the detection of spuriously high mutation rates at this site[49,50].

**Total RNA-associated protein purification (TRAPP)**. K562 were grown for 10 divisions in SILAC RPMI (Thermo Fischer; cat. 88365) supplemented with 10% dialyzed FBS (Gibco; cat. 26400044) and 50 µg/L each of lysine and arginine. For "light" cultures, these amino acids were obtained from Sigma. For "heavy" cultures, $^{13}C_6$-lysine and $^{13}C_6$-arginine were obtained from Cambridge Isotope Laboratories (cat. CLM-226 and CLM-2247, respectively).

Cells were grown to a density of $0.5–0.8 \times 10^6$ cells/mL, with minimum 90% viability as determined by Trypan blue exclusion (Invitrogen; cat. T10282) using a Countess automated cell counter. 25 mL aliquots of culture were transferred to a custom quartz dish, and crosslinked with 400 mJ/cm$^2$ of UVC using a Vari-X-Link device[61]. Enough aliquots were crosslinked to create a sample containing $50 \times 10^6$ cells. After cross-linking, cells were pelleted, washed with ice-cold PBS, and frozen at −80 °C.

RNA-associated proteins were purified following a modified version of the published TRAPP protocol, where silica columns were used in place of silica beads[37]. Proteins were digested on the column with 0.25 µg of Trypsin/Lys-C protease mix (Promega; cat. V5071), and peptides eluted for mass spectrometry. Samples were acquired by the Proteomics Facility at the Wellcome Centre for Cell Biology, University of Edinburgh, on an Orbitrap Fusion Lumos Tribrid Mass Spectrometer (Thermo Fisher Scientific, UK). Raw data were processed by the MaxQuant software platform, version 1.6.1, searching against the UniProt reference proteome set. Further analysis used custom scripts. A paper describing this modified TRAPP protocol is in preparation.

**Proteomics**. Cells were lysed in 1x Laemmli buffer, quantified, and boiled for 5 min. 20 µg of protein was run per lane on a 12% pre-cast gel (BioRad; cat 4561043), with size determined using a pre-stained protein ladder run in lane 1 (NEB; cat. P7719S). The gel was washed three times with MQ, each for 5 min with nutation, then stained with Imperial protein stain (Thermo Scientific; cat. 24615) for 1 hour, and destained with MQ overnight. The gel was cut into fractions, and each fraction dissected into cubes not more than 1 mm$^3$ in volume. Proteins were then digested with trypsin and StageTips created, following a published protocol[62]. Samples were acquired and analyzed as described for TRAPP.

**Cross-linking and analysis of cDNAs (CRAC)**
*Preparation of human CRAC samples.* Cells were grown and crosslinked as for TRAPP, except that normal RPMI media was used rather than SILAC media.

*Preparation of samples for yeast Pop1 CRAC.* 2.75 L of yeast culture was grown to 0.5 OD, and irradiated for 100 s with UVC (254 nM) in a custom "Megatron" cross-linking device, as described[47]. Cells were then pelleted (2700 RCF for 15 min at 4 °C) and washed with ice-cold PBS, pelleted again, and frozen at −80 °C until use.

*CRAC protocol*. For yeast Pop1, CRAC was performed following a published protocol[63]. For human CRAC, technical modifications were made to this protocol. The full human CRAC protocol is described below.

*Sequencing and analysis of CRAC data*. CRAC libraries were sequenced either on a MiniSeq or HiSeq, both with 150 base reads. Yeast CRAC data were processed using custom scripts calling utilities from the PyCRAC collection[64]. First, raw FASTQ files were demultiplexed using pyBarcodeFilter. Then, adapters and low-quality sequences were removed with Flexbar (version. 3.4.0)[65]. Next, PCR duplicates were collapsed with pyFastqDuplicateRemover. Reads were then aligned to the *Saccharomyces cerevisiae* genome sequence from Ensembl release EF4.74 [ftp://ftp.ensembl.org/pub/release-74/gtf/saccharomyces_cerevisiae] by Novoalign version 2.07.00. Read counts were produced using pyReadCounters, and pileups with pyPileup. Graphs were produced with custom Python3 scripts, available on request. For human data, low-quality reads were removed with Flexbar. STAR (version. 2.7.3a) was then used to align reads to a custom transcriptome database, based on GRCh38.p13 [https://www.ncbi.nlm.nih.gov/assembly/GCF_000001405.39] with additional tRNA and rRNA species included. Read counts and pileups were generated with custom scripts, available on request.

**CRAC protocol for human cell samples**. Buffers were prepared as in Supplementary Table 8. Crosslinked cell pellets were resuspended in 3 mL of ice-cold buffer LB, supplemented with cOmplete EDTA-free Protease Inhibitor Cocktail (Roche; cat. 11873580001; 1 tablet per 50 mL lysis buffer), and left to lyse for 10 min on ice. To ensure complete lysis, 0.5 mL of zirconium oxide beads (Thistle Scientific; cat. ZrOB05) were then added and the samples vortexed for 30 s. Samples were then centrifuged at 1000 g for 5 min at 4 °C, and the supernatant clarified by syringe-filtration through a 0.22 μM filter.

*anti-FLAG IP*. 100 μL of anti-FLAG M2 magnetic beads (Milipore; cat. M8823) were washed twice with 0.5 mL of buffer LB, then added to the sample and nutated at 4 °C for 2 h. Using a magnetic rack, beads were then washed three times with 1 mL buffer LB and twice with 1 mL buffer C, then resuspended in 300 μL of buffer C.

*Partial RNA digestion*. RNA was partially digested by adding 300 μL of Buffer C supplemented with 0.04 units of RNace-IT (Agilent; cat. 400720) and incubating at 23 °C for 10 min with shaking (100 RPM). Beads were then washed once with 0.75 mL of buffer LB, once with 0.75 mL of buffer FA2 (3 min on ice with nutation), once with 0.75 mL of buffer FA3 (again, on ice with nutation), and, finally, twice with 0.75 mL of buffer LB.

*Elution from FLAG beads*. Buffer was removed from the beads using a magnetic rack. The beads were then resuspended in 200 μL of buffer LB supplemented with 150 μg / mL of 3x FLAG peptide (Sigma; cat. F4799) and incubated for 5 min at 37 °C, shaking at 1200 RPM. The eluate was collected and the elution repeated. The two eluates were pooled and 800 μL of buffer WB1 added to denature proteins.

*HIS-tag purification*. 75 μL of nickel beads (Ni-NTA Aragose; Qiagen; cat. 30210) were washed twice with 1 mL of buffer WB1, spinning at 1000 RCF for 30 seconds and removing supernatant between washes. Eluate was added to the beads and incubated for 2 h at RT with nutation. Beads were then washed as before, three times with 1 mL of buffer WB1 and twice with 0.75 mL of buffer C. After the last wash, beads were resuspended in 600 μL of buffer C and transferred to a SigmaPrep column (Sigma; cat. SC1000).

*RNA dephosphorylation*. Beads on the column were washed twice with 0.75 mL of 50 mM Bis-Tris pH 6.5. For this and all column steps, buffer was allowed to pass through the column using gravity flow. Dephosphorylation was then initiated by adding 80 μL of reaction mix containing 4 μL of T4 Polynucleotide Kinase (NEB; cat. M0201L), 2 μL of TSAP Thermosensitive Alkaline Phosphatase (Promega; cat. M9910) and 2 μL of RNasin Ribonuclease Inhibitor (Promega; cat. N2511), in 1x R2 buffer (diluted from 10x stock with MQ). Samples were incubated for 30 min at 37 °C, and then the columns were washed once with 0.5 mL of buffer WB1 and three times with 0.75 mL of buffer C.

*3′ Linker ligation*. To facilitate cloning, a DNA linker was ligated to the 3′ of the RNA. This linker was synthesised by IDT. It has a blocked 3′ end and an activated adenosine at the 5′ end, with sequence: 5′-rAppTGGAATTCTCGGGTGCCAA GG/ddC/-3′. Ligation was initiated by adding 80 μL of reaction mix containing 8 μL of 10 μM linker, 4 μL of truncated T4 RNA ligase II (NEB; cat. M0373L), 2 μL of RNasin and 20 μL of 50% PEG8000, in 1x R1 buffer. The reaction was left to proceed overnight at 16 °C. The next morning, columns were washed once with 0.5 mL of buffer WB1, once with 0.75 mL of buffer B2, once with 0.5 mL of buffer B3, once with 0.75 mL of buffer B2, and, finally, three times with 0.75 mL of buffer C.

*RNA labelling, 5′ linker ligation and elution*. RNA was radioactively labelled by adding 80 μL of reaction mix containing 4 μL of $^{32}$P-γATP (10 μCi/μL; Hartmann

Analytic), 2 μL of T4 Polynucleotide Kinase, 2 μL of RNasin and 20 μL of 50% PEG8000, in 1x R1 buffer. After 30 min at 37 °C, 1 μL of 100 mM ATP (ThermoFisher; cat. 10304340) was added, and the reaction left for an additional 15 min. A 5′ linker was then ligated by adding 4 μL of T4 RNA ligase 1 (NEB; cat. M0204L) and 4 μL of 100 μM barcoded linker. The reaction was performed for 3 h at 16 °C then 2 h at 25 °C. The 5′ linker was again produced by IDT. It has an inverted ddT at the 5′ end to increase stability and a barcode sequence at the 3′ end. Also at the 3′ end, there is a random 3-mer to allow collapsing of PCR duplicates. Following ligation, columns were washed three times with 0.5 mL of buffer WB1. RNA:protein complexes were then eluted by incubating beads with 50 μL of buffer BE (5 min at RT, shaking at 800 RPM), then spinning out eluate (1000 RCF for 1 min at RT). Elution was repeated, and eluates combined. RNA:protein complexes were then precipitated by adding 9 volumes of 100% ethanol and 3 μL of GlycoBlue co-precipitant (Invitrogen; cat. AM9516), and incubating samples overnight at −20 °C.

*Size selection of RNA:protein complexes*. The next morning, samples were centrifuged at 21,000 RCF for 20 min at 4 °C and the supernatant removed. Pellets were washed twice with 1 mL of 80% acetone, then allowed to dry. Pellets were then resuspended in 12 μL of MQ and left to rehydrate for 10 min at RT. 4 μL of 4x NuPage LDS sample buffer (Invitrogen; cat. NP0007) was added, together with 2-Mercaptoethanol to a final concentration of 2%.

Samples were then denatured at 65 °C for 10 min, and loaded onto a NuPage 4-12% polyacrylamide gel (Invitrogen; cat. NP0321). The gel was run at 120 V in 1x NuPAGE MOPS SDS Running Buffer (Invitrogen; cat. NP0001), then transferred onto Hybond-CExtra membrane (Fischer Scientific; cat. 10564755) in 1x NuPAGE transfer buffer (Invitrogen; cat. NP0006) supplemented with 10% methanol, for 2 h at 100 V.

Membranes were then exposed overnight to Kodak BioMax MS autoradiography film (Sigma; cat. K8222648). Using the radiograph as a guide, areas corresponding to RNA libraries were cut from the membrane, and proteins digested in buffer PKB supplemented with 100 μg of proteinase K (Sigma; cat. 3115887001) for 1 hour at 55 °C with shaking (1400 RPM). RNA was then extracted by adding 50 μL of 3 M NaAC and 50 μL of buffer PCI, centrifuging at 20,000 RCF for 5 min at 4 °C, and taking the upper phase into a new tube. RNA was then precipitated by adding 1 mL of 100% ethanol and 2 μL of GlycoBlue, and incubating overnight at −20 °C.

*Reverse transcription*. The following morning, samples were centrifuged at 21,000 RCF (4 °C for 20 min). The supernatant was removed and pellets washed with 70% ethanol, before being allowed to dry. The pellet was resuspended in 11 μL of MQ supplemented with 1 μL of 10 μM miRCat RT oligo (sequence: CCTTGGCACCC GAGAATT) and 2.5 μL of 2.5 mM dNTP mix (diluted from 100 mM stocks: Invitrogen; cat. 10297117). The mix was heated to 80 °C for 3 min then chilled on ice for 5 min. 6 μL of RT mix was then added, composed of 4 μL SuperScript IV RT Reaction Buffer (Invitrogen; cat 18090050B), 1 μL of 100 mM DTT and 1 μL RNasin. The mix was heated to 50 °C for 5 min, then 1 μL of Superscript IV reverse transcriptase (Invitrogen; cat 18090010) was added and the reaction left for 15 min. The mixture was then incubated at RT for 5 min, then on ice for 3 min. 2 μL of Exonuclease I (NEB; cat. M0293S) was then added and the mixture incubated for 30 min at 37 °C, before heating to 80 °C for 20 min to deactivate the enzyme.

*Library preparation*. DNA libraries were amplified using the primers P5 Forward (AATGATACGGCGACCACCGAGATCTACACTCTTTCCCTACACGACGCTC TTCCGATCT) and PE miRCat Reverse (CAAGCAGAAGACGGCATACGAG ATCGGTCTCGGCATTCCTGGCCTTGGCACCCGAGAATTCC). PCR reactions were assembled in 50 μL and contained 0.3 μL of Phusion High-Fidelity DNA Polymerase (NEB; cat. M0530S), 10 μL of provided 5x Phusion buffer, 5 μL of 2.5 mM dNTP mix, 1 μL of each primer, and 2 μL of cDNA. cDNA was amplified using the following programme: initial denaturation at 98 °C for 30 s, then 18 to 22 cycles of 98 °C for 10 s, 65 °C for 30 s and 72 °C for 30 s. The final extension was at 72 °C for 5 min. Three reactions were completed for each sample. Once complete, the PCR reactions were pooled for each sample and purified using AMPure XP beads (Beckman Coulter; cat. A63880). Per reaction mix, 2 volumes of AMPure buffer and 0.2 volumes of AMPure XP beads were added. Samples were mixed and incubated for 5 min at RT. The supernatant was then removed on a magnetic rack, and beads washed twice with 200 μL of 80% ethanol. After the last wash, ethanol was removed and the beads left to dry.

Purified DNA was then eluted by resuspending the beads in 10 μL of MQ, and 2 μL of 6x DNA gel loading dye added (Thermo Scientific; cat. R0611). A 3% Metaphor agarose (Lonza; cat. 50181) gel was prepared in 1x TBE. Samples were loaded and run at 80 V until dye reached the bottom of the gel. The smear corresponding to the DNA library was extracted with a Zymoclean Gel DNA Recovery Kit (Zymo; cat. D4007), and fluorometrically quantified. Libraries were diluted to approximately 5 nM and pooled, and submitted for sequencing on an Illumina HiSeq2500 platform at the Wellcome Trust Clinical Research Facility, University of Edinburgh.

**Statistics**. All statistics were calculated using Prism 9.0.0, with tests as indicated in figure legends. In general, plots show mean values + / − standard deviations.

**Reporting Summary**. Further information on research design is available in the Nature Research Reporting Summary linked to this article.

## Data availability

The data that support this study are available from the corresponding author upon reasonable request. The GEO accession number for all sequence data reported in this paper is GSE171021. The proteomics data are available through the ProteomeXchange partner repository with the dataset identifier PXD025029. Human reference sequences used in this paper were taken from GRCh38.p13 [https://www.ncbi.nlm.nih.gov/assembly/GCF_000001405.39]. Yeast sequences are from the *Saccharomyces cerevisiae* genome (Ensembl release EF4.74 [ftp://ftp.ensembl.org/pub/release-74/gtf/saccharomyces_cerevisiae]). Mouse sequences are from GRCm39 [https://www.ncbi.nlm.nih.gov/assembly/GCF_000001635.27/]. Source data are provided with this paper.

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

## Acknowledgements

We thank Prof. Sophie Hambleton from Newcastle University for providing patient and control fibroblasts, sourced from the Great North Biobank (REC number 16-NE-0002). We also thank Richard Clark, Martin Waterfall and Fiona Rossi for technical assistance, and Sarah Walmsley and Edward Wallace for helpful discussions. This work was supported by Wellcome through an Edinburgh Clinical Academic Track fellowship to NR (213011), a Wellcome Principal Research Fellowship to DT (077248) and Wellcome Trust Investigator Award to RZ (WT205014/Z/16/Z). Work in the Wellcome Centre for Cell Biology is supported by a Centre Core grant (203149).

## Author contributions

N.R., R.Z., and D.T. conceived the project and wrote the manuscript. N.R., V.S., D.W., T.T., and C.S. performed experiments. N.R., V.S., D.W., T.T., A.H., R.Z., and D.T. designed experiments. N.R., V.S., C.S., and T.T. analyzed data. All authors edited and reviewed the manuscript.

## Competing interests

The authors declare no competing interests.
