## [Peer Review File · Nature Communications]

A disease-linked lncRNA mutation in RNase MRP inhibits ribosome synthesisREVIEWER COMMENTS

Reviewer #1 (Remarks to the Author):

Robertson et al 2021

Previous work by Goldfarb (Genes Dev 2017) used CRISPR-Cas9 editing to eliminate the endogenous human RMRP RNA locus that is mutated in cartilage Hair Hypoplasia (CHH). This study established a role for RMRP RNA in human pre-ribosomal RNA processing.

In the current work, the authors validate this previous work and extend it to show that mutations in RMRP impair mouse T cell activation and re-rRNA processing. They further describe rRNA processing defects and growth arrest in CHH patient-derived fibroblasts and recapitulate the rRNA processing defect in human cells engineered to carry a disease-linked RMRP mutation by CRISPR targeting. The mutant human cells show impaired pre-rRNA processing, reduced mature rRNA and a reduced ratio of cytoplasmic to mitochondrial ribosomes. The 70AG mutations reduce the levels of intact RNase MRP complexes likely due to destabilization of the interaction between the POP1 proteins and RMRP. The paper is clearly written, the approaches taken are elegant and convincing and together, the data support the conclusions that the authors wish to draw. The manuscript provides new insight into the pathogenesis of CHH, supporting its definition as a ribosomopathy, and will be of interest to the broad ribosome assembly community. Statistical analysis seems appropriate and valid.

I have minor comments only.

1. For the section of the paper describing the CRAC results it would be helpful for the general reader to include a figure showing the architecture of the yeast RMRP complex to help visualize the system.
2. P6: Please explain in the text the rationale for using OT-1 Rag^{-/-} mice?
3. P7: text says, "Four independent, homozygous CRISPR clones were obtained (Fig. S3B)". However, Figure S3B only shows data for 3 clones.

4. The authors raise the intriguing question of why the phenotypes of DBA and CHH are different. Is there any evidence of induction of p53 stabilization in the primary T cells engineered to carry RMRP mutations? Is p53 activation described in hematopoietic cells from patients with CHH?

5. The cited review by Narla et al from 2010 is great but is now quite outdated. Suggest replacing with a more recent summary.

Reviewer #2 (Remarks to the Author):

In this manuscript, the authors describe the mechanism by which mutant RMRP inhibits ribosome biogenesis. Briefly, RMRP is a non-coding RNA that is the core of the RNase MRP complex, and mutations in RMRP are the causative agent of Cartilage Hair Hypoplasia (CHH), a ribosomopathy. Further, the 70AG mutant (most common variant for CHH), was shown by the authors to decrease pre-rRNA processing, decrease mature rRNA concentrations, and resulted in a decreased ratio of cytosolic to mitochondrial ribosomes, as well as a reduction in RNase MRP complexes. The authors conclude that CHH is the first processing-specific ribosomopathy.

This last point is contentious as other ribosomopathies have been described as resulting from defects in the processing of the pre-rRNA (Freed et al. 2010 NAR; Farrar et al., 2014, Am J Hematol ; McCann et al 2016, eLife; Bryant et al., 2021, PNAS). This list of the prior literature is not exhaustive. The authors should be aware that this manuscript certainly does not report the first processing-specific ribosomopathy, and they should not claim so. Perhaps the authors mean that this is the first ribosomopathy to be caused by a mutant ncRNA and not the first processing-specific ribosomopathy? I don't think that is true either, as variants in the U8 snoRNA cause leukoencephalopathy with calcifications and cysts. I suggest taking all of these claims out, and letting the work stand for itself. It is not the "first" in any of these categories.

An excellent, scholarly reference for ribosomopathies is Table 1 in Warren 2017.

The authors do reference Thiel et al., 2005 in the Results section, which previously demonstrated a defect of pre-rRNA processing (another example of a ribosomopathy with a processing defect, this time

in the same disease) and a depletion of the 5.8S rRNA species in CHH patient fibroblast cells. It should also be included in the Discussion with an analysis of how this new work extends the prior work and adds to it to make it novel.

The following points are comments made throughout the manuscript.

I. In lines 60-61, a reference to the previously found role in ribosome biogenesis for RMRP should be cited.

II. In line 67, the authors should include that the 47S is transcribed by RNA Polymerase I and in lines 69-70, it should include that the 5S is transcribed by RNA Polymerase III.

III. In lines 72-73, the authors should be more explicit in that their results do show a depletion of cytosolic ribosomes.

IV. In line 151, perhaps include a reason for the decreased efficiency.

V. In lines 174-175, include a reference to where these guide sites are and if they are relevant to the CHH mutant site (as this is referenced in lines 207-209 for the K562 cells).

VI. In line 201, it would be helpful to include literature relevant references regarding the preferential use of one processing pathway over another.

VII. In line 213: is there any information on how the ncRNA RPPH1 is affected/involved in CHH?

VIII. For Figure 2, since a comparison is made to the previously published literature in lines 230-237, it would be good to perform qPCR for a more direct comparison (in addition to the performed northern blot analysis).

IX. In the discussion of Figure 4, do we know these mitochondrial ribosomes are functional in mutant RMRP cells?

X. In line 296, there appears to be a typo—should it be “analyses of total protein”?

XI. For Figure 5, specifically in reference to lines 310-313, can the authors elaborate on why the decrease would not be statistically significant?

XII. Is the interaction site between RMRP and POP1 relevant to the disease mutation site?

Reviewer #3 (Remarks to the Author):

The assembly pathway of ribosomes is a core cellular process in all organisms. A number of human diseases (termed ribosomopathies) have been described that are caused by mutations in genes

encoding ribosomal proteins or ribosome assembly factors. In this manuscript, Roberston and colleagues analyse the effects of the most common mutation in the non-coding RNA component of the RNase MRP that is linked to Cartilage Hair Hypoplasia (CHH), which is a disorder characterised by dwarfism, sparse hypoplastic hair, defective T cell function and increased risk of malignancies. They show that disruption of the RNA component of RNase MRP (RMRP) by transfection of CRISPR guide RNAs targeting this gene in mouse T cells affects ribosomal RNA precursor processing and T cell function. More specifically, the 70AG mutation, which is most common in CHH patients, reduces cell proliferation in human fibroblasts, impairs RNase MRP assembly and pre-rRNA processing, leading to accumulation of the 41S pre-rRNA intermediate and reduced levels of cytoplasmic ribosomes. In contrast to a previous report suggesting a change in the ratio of the 5.8SS to 5.8SL rRNA variants in patient cells compared to healthy individuals, the authors of this study do not observe such effect of the 70AG mutation and suggest that this is due to the detection method (qPCR) used for the previous publication, which seems plausible.

This study is well performed and represents the first thorough analysis of the effects of the major CHH mutation on RNase MRP function in ribosome synthesis. While indications that the 70AG mutation might affect ribosome synthesis were previously observed (Thiel et al 2005, Steinbusch et al 2017) the current study clearly documents that CHH constitutes a novel ribosomopathy. However, to merit publication in Nature Communications the following major points need to be addressed:

1. A more thorough investigation of the effects of the 70AG mutation on the assembly of RNase MRP would increase the novelty of this report. Besides the analysis of protein binding possible effects of the mutation of RNA structure or solvent accessibility should be analysed by SHAPE or a similar method. As the mutation has such strong effects on RNase MRP function, ribosome assembly and therefore cellular proliferation, more significant effects can be expected on assembly, structure and/or function of the RNP. Besides (potentially less likely) direct effects of the mutation, RNP structure could also be affected by lack of/changes in protein binding.
2. The effects on RNase MRP assembly and structure in 70AG mutant cells will be a strong aspect of the manuscript. The already existing and the new data on this topic should be discussed more thoroughly and also related to the structural information and RNP interactions available on the yeast homologues.
3. The manuscript is written in a very compact manner. To make the contents accessible to the broad readership of Nature Communications, it is essential that the introduction (e.g. on human ribosome assembly) is significantly expanded and the results are presented and discussed in a more accessible manner. This should make the manuscript accessible also to clinicians and people who are not directly from the field, so it becomes a go-to resource for CHH as a ribosomopathy.

Minor points:

1. The nucleotide positions should be indicated in Fig. S3A.

2. Also, the paper should include a 2D structure of the RNA with the crosslinking sites identified by CRAC indicated with a colour gradient (to indicate peak height). Do the suggested interactions of POP1 and POP4 overlap and what do the authors make of this?

REVIEWER COMMENTS

We thank the referees for their careful consideration and helpful comments, which we feel have improved the MS.

Reviewer #1

1. For the section of the paper describing the CRAC results it would be helpful for the general reader to include a figure showing the architecture of the yeast RMRP complex to help visualize the system.

An explanatory schematic has been added as Fig. S5.

2. P6: Please explain in the text the rationale for using OT-1 Rag^{-/-} mice?

New sentence added: "This system allows T cells to be activated in a relatively physiological manner, by adding Ova peptide to the culture medium."

Rag is deleted in these mice to prevent Rag-mediated rearrangement of the Ova-specific TCR allele.

3. P7: text says, "Four independent, homozygous CRISPR clones were obtained (Fig. S3B)". However, Figure S3B only shows data for 3 clones.

4 independent clones were obtained and confirmed by Sanger sequencing. However, for practical reasons, not all experiments were performed using all 4 clones. Clones used are indicated in Figure legends.

4. The authors raise the intriguing question of why the phenotypes of DBA and CHH are different. Is there any evidence of induction of p53 stabilization in the primary T cells engineered to carry RMRP mutations? Is p53 activation described in hematopoietic cells from patients with CHH?

We did not assay p53 in our T cell CRISPR experiments, although this would be an interesting experiment to pursue in the future. We are not aware of any data directly assessing p53 status cells from CHH patients. However, T cells from these patients do have increased activation-induced cell death, which would be consistent with p53 stabilization (de

la Fuente et al. 2015, J Allergy Clin Immunol). Furthermore, zebrafish with a CRISPR-generated 13-bp deletion in a conserved region of *rmrp* show increased apoptosis associated with elevated expression of *tp53* (Sun et al. 2019, J Bone Miner Res). Recently a direct role for *RMRP* in regulating p53 was reported. Chen *et al* showed that *RMRP* retains SNRPA1 (small nuclear ribonucleoprotein polypeptide A') in the nucleus, which in turn causes p53 degradation via MDM2 (Chen et al. 2021, PNAS).

5. The cited review by Narla et al from 2010 is great but is now quite outdated. Suggest replacing with a more recent summary.

The references have been updated to also include Warren (2017, 2018) and Farley-Barnes et al. (2019). We also now explicitly introduce ribosomopathies in the Introduction.

Reviewer #2

The authors conclude that CHH is the first processing-specific ribosomopathy. This last point is contentious as other ribosomopathies have been described as resulting from defects in the processing of the pre-rRNA (Freed et al. 2010 NAR; Farrar et al., 2014, Am J Hematol ; McCann et al 2016, eLife; Bryant et al., 2021, PNAS). This list of the prior literature is not exhaustive. The authors should be aware that this manuscript certainly does not report the first processing-specific ribosomopathy, and they should not claim so. Perhaps the authors mean that this is the first ribosomopathy to be caused by a mutant ncRNA and not the first processing-specific ribosomopathy? I don't think that is true either, as variants in the U8 snoRNA cause leukoencephalopathy with calcifications and cysts. I suggest taking all of these claims out, and letting the work stand for itself. It is not the "first" in any of these categories.

The abstract has been altered to remove these points, which were indeed somewhat overstated.

An excellent, scholarly reference for ribosomopathies is Table 1 in Warren 2017.

The reference has been included in the Introduction.

The authors do reference Thiel et al., 2005 in the Results section, which previously demonstrated a defect of pre-rRNA processing (another example of a ribosomopathy with a processing defect, this time in the same disease) and a depletion of the 5.8S rRNA species

in CHH patient fibroblast cells. It should also be included in the Discussion with an analysis of how this new work extends the prior work and adds to it to make it novel.

How our work extends previous analyses has been more explicitly stated in the Discussion. We would point out that, while Thiel et al., 2005 reported fine work, they utilized only two Taqman PCR probes in analyses of ribosome synthesis. These were designed based on a prediction that MRP cleavage would take place at the 5' end of 5.8S rRNA, which does not match our much more detailed analyses. The conclusion of 5.8S depletion is based on the change in the ratio of a probe across the 5.8S-ITS1 boundary, relative to a probe within mature 5.8S. Absolute abundances for 5.8S or other rRNAs were not assessed.

The following points are comments made throughout the manuscript.

I. In lines 60-61, a reference to the previously found role in ribosome biogenesis for RMRP should be cited.

Introduction to the previous work has been clarified and highlighted.

II. In line 67, the authors should include that the 47S is transcribed by RNA Polymerase I and in lines 69-70, it should include that the 5S is transcribed by RNA Polymerase III.

Included in the revised text.

III. In lines 72-73, the authors should be more explicit in that their results do show a depletion of cytosolic ribosomes.

Changed in revised text.

IV. In line 151, perhaps include a reason for the decreased efficiency.

We have added a new sentence to clarify this point: "Guide efficacy in this assay is predicted to depend on both guide cutting rates and the tolerance to small mutations within the target region in the RNA."

V. In lines 174-175, include a reference to where these guide sites are and if they are relevant to the CHH mutant site (as this is referenced in lines 207-209 for the K562 cells).

Guide positions are shown schematically in a new figure, Fig. S2A.

VI. In line 201, it would be helpful to include literature relevant references regarding the preferential use of one processing pathway over another.

Although there are mutations that favour one or other pathway, the physiological significance of the alternative pathways remains unclear - even in yeast where this has previously been actively investigated. It may be that the presence of partially redundant pathways enhances the overall efficiency and resilience of the system. We mention this in the revised Introduction.

VII. In line 213: is there any information on how the ncRNA RPPH1 is affected/involved in CHH?

To the best of our knowledge, there is no involvement of RPPH1 in CHH induced by mutation of RMRP. Reported mutations in *POP1* are very rare, and linked to related but distinct diseases including Anauxetic dysplasia, which might be caused by impaired function of RNase P plus MRP.

VIII. For Figure 2, since a comparison is made to the previously published literature in lines 230-237, it would be good to perform qPCR for a more direct comparison (in addition to the performed northern blot analysis).

We have performed this analysis and include it as Figure S3E. The RT-PCR confirms accumulation of pre-rRNA species including ITS1 and ITS2 relative to 47S (as determined using the 5' ETS), consistent with the northern data.

IX. In the discussion of Figure 4, do we know these mitochondrial ribosomes are functional in mutant RMRP cells?

We have no data on the functionality of the mitochondrial ribosomes in RMRP mutant cells. However, CHH patients do not exhibit features linked to other mitochondrial diseases, so we predict that they are functional.

X. In line 296, there appears to be a typo—should it be “analyses of total protein”?

Wording has been updated as suggested.

XI. For Figure 5, specifically in reference to lines 310-313, can the authors elaborate on why it the decrease would not be statistically significant?

In qPCR assays of *RMRP* in wildtype and 70^{AG} cells, there were generally small changes which were inconsistent in direction between replicates. Combining multiple independent experiments showed no significant difference (Fig. S3C).

XII. Is the interaction site between RMRP and POP1 relevant to the disease mutation site?

Site 201 is not currently known to be linked to the site of the 70^{AG} mutation. We have added a new schematic figure, Fig. 5E, showing how this cross-link site relates to the disease mutation site and POP4 cross-linking sites.

Reviewer #3

1. A more thorough investigation of the effects of the 70AG mutation on the assembly of RNase MRP would increase the novelty of this report. Besides the analysis of protein binding possible effects of the mutation of RNA structure or solvent accessibility should be analysed by SHAPE or a similar method. As the mutation has such strong effects on RNase MRP function, ribosome assembly and therefore cellular proliferation, more significant effects can be expected on assembly, structure and/or function of the RNP. Besides (potentially less likely) direct effects of the mutation, RNP structure could also be affected by lack of/changes in protein binding.

We have investigated potential changes in RNA structure using in-cell SHAPE-MaP, and added this data to Fig. 5E, and new Figs. S8 and S9. In general, the SHAPE reactivity profiles obtained for wildtype and 70AG cells are similar, presumably reflecting the subtle nature of survivable mutations. Applying statistical analysis to three independent pairs of wildtype/70AG profiles reveals two areas where the mutant cells show reproducibly increased reactivity, one of which is just downstream of the mutation site (nucleotides 73 and 74; Fig. S9). The other is nucleotides 106 and 107. We are cautious about interpreting this data given the generally small and inconsistent changes between replicates (Fig. S8). However, we conclude that the 70AG mutation does not cause significant refolding of the *RMRP* RNA in-cell.

2. The effects on RNase MRP assembly and structure in 70AG mutant cells will be a strong aspect of the manuscript. The already existing and the new data on this topic should be discussed more thoroughly and also related to the structural information and RNP interactions available on the yeast homologues.

We have included further discussion of this point. We also present additional data on the interactions within yeast RNase MRP, based on *in vivo* crosslinking of Pop1, as Figure S7B. This highlights apparent differences between the complexes present in yeast and humans.

3. The manuscript is written in a very compact manner. To make the contents accessible to the broad readership of Nature Communications, it is essential that the introduction (e.g. on human ribosome assembly) is significantly expanded and the results are presented and discussed in a more accessible manner. This should make the manuscript accessible also to clinicians and people who are not directly from the field, so it becomes a go-to resource for CHH as a ribosomopathy.

Minor points:

1. The nucleotide positions should be indicated in Fig. S3A.

Nucleotide positions have been indicated at 50 base intervals.

2. Also, the paper should include a 2D structure of the RNA with the crosslinking sites identified by CRAC indicated with a colour gradient (to indicate peak height). Do the suggested interactions of POP1 and POP4 overlap and what do the authors make of this?

As requested, a 2D structure with crosslinking sites indicated has been included as Figure 5E. Relative extent of peak heights are indicated in the legend rather than with a colour gradient to enhance readability.

There are indeed some overlaps in the recovered crosslinking sites. This has previously been seen for other RNPs (see for example Hunziker et al. (2016) Nat. Commun., 7, 12090. PMID: PMC4931317). Our interpretation is that *in vivo* RNPs can show flexibility, perhaps on different substrates or at different steps during their function. We mention this point in the revised text (P12).

REVIEWERS' COMMENTS

Reviewer #1 (Remarks to the Author):

The authors have addressed all the points raised by this reviewer. They have suitably revised and further improved their manuscript. This is a high quality and very interesting study.

Reviewer #2 (Remarks to the Author):

None further

Reviewer #3 (Remarks to the Author):

The authors have addressed the concerns and comments of the reviewers and in my opinion the manuscript can now be published in Nature Communications.